# MMWorld: Towards Multi-discipline Multi-faceted Video Understanding Evaluation

**Xuehai He[1], Weixi Feng[2], Kaizhi Zheng[1], Yujie Lu[2], Wanrong Zhu[2], Jiachen Li[2], Yue Fan[2], Jianfeng Wang[3], Linjie Li[3], Zhengyuan Yang[3], Kevin Lin[3], William Yang Wang[2], Lijuan Wang[3], Xin Eric Wang[1]**

[1]UCSC, [2]UCSB, [3]Microsoft
**Correspondence:** xhe89,xwang366@ucsc.edu

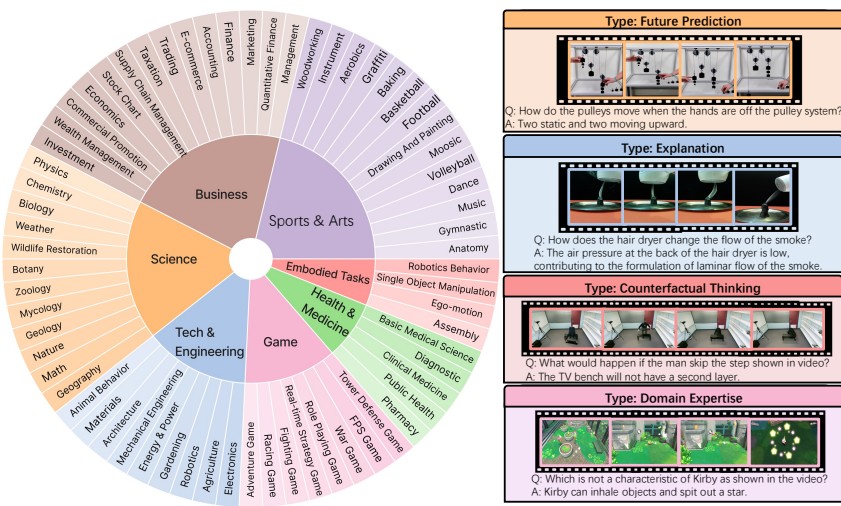

Figure 1: MMWorld covers seven broad disciplines and 69 subdisciplines, focusing on the evaluation of multi-faceted reasoning beyond perception (e.g., explanation, counterfactual thinking, future prediction, domain expertise). On the right are four video samples from the Science, Tech & Engineering, Embodied Tasks, and Game disciplines.

## Abstract

Multimodal Language Language Models (MLLMs) demonstrate the emerging abilities of "world models"—interpreting and reasoning about complex real-world dynamics. To assess these abilities, we posit videos are the ideal medium, as they encapsulate rich representations of real-world dynamics and causalities. To this end, we introduce MMWorld, a new benchmark for multi-discipline, multi-faceted multimodal video understanding. MMWorld distinguishes itself from previous video understanding benchmarks with two unique advantages: (1) **multi-discipline**, covering various disciplines that often require domain expertise for comprehensive understanding; (2) **multi-faceted reasoning**, including explanation, counterfactual thinking, future prediction, etc. MMWorld consists of a human-annotated dataset to evaluate MLLMs with questions about the whole videos and a synthetic dataset to analyze MLLMs within a single modality of perception. Together, MMWorld encompasses 1,910 videos across seven broad disciplines and 69 subdisciplines, complete with 6,627 question-answer pairs and associated captions. The evaluation includes 4 proprietary and 11 open-source MLLMs, which struggle on MMWorld (e.g., GPT-4o performs the best with only 62.5% accuracy), showing large room for improvement. Further ablation studies reveal other interesting findings such as models' different skill sets from humans. We hope MMWorld can serve as an essential step towards world model evaluation in videos.

# 1 INTRODUCTION

Foundation models, such as Large Language Models (LLMs) (OpenAI, 2023c; Touvron et al., 2023a; Jiang et al., 2023; Anil et al., 2023) and Multimodal LLMs (MLLMs) (Team et al., 2023; Lin et al., 2023a; Li et al., 2023c; Maaz et al., 2024; Chen et al., 2023), have demonstrated remarkable abilities in text and image domains, igniting debates about their potential pathways to Artificial General Intelligence (AGI). This raises a critical question: how well do these models understand the dynamics of the real world? Are they equipped with an inherent World Model (LeCun, 2022; Chen et al., 2024; Ha & Schmidhuber, 2018; Xiang et al., 2024) that can understand and reason about the underlying principles and causalities of the dynamic, multimodal world?

Videos, with their rich, dynamic portrayal of the real world, are ideally suited for evaluating the "world modeling" capabilities of MLLMs. Existing video understanding benchmarks (Li et al., 2023d; Ning et al., 2023b; Pătrăucean et al., 2023; Li et al., 2023d), however, fall short in two key perspectives for such evaluations. First, as LeCun et al. (LeCun, 2022) discussed, the world model should be able to *(1) estimate missing information about the state of the world not provided by perception, and (2) predict plausible future states of the world*. Evaluation of such capabilities requires **multi-faceted reasoning** beyond perception level, including explaining the video dynamics, counterfactual thinking of alternative consequences, and predicting future activities within videos. Moreover, the **multi-discipline** nature of the multimodal world necessitates a grasp of diverse fundamental principles—ranging from physics and chemistry to engineering and business. Hence, domain expertise across a variety of disciplines is imperative for a thorough evaluation of a model's world understanding towards AGI (Morris et al., 2023; Yue et al., 2023).

Therefore, we introduce MMWorld, a multi-discipline multi-faceted multimodal video understanding benchmark to comprehensively evaluate MLLMs' abilities in reasoning and interpreting real-world dynamics [1]. MMWorld encompasses a wide range of disciplines and presents multi-faceted reasoning challenges that demand a combination of visual, auditory, and temporal understanding. It consists of 1,910 videos that span seven common disciplines, including *Art & Sports*, *Business*, *Science*, *Health & Medicine*, *Embodied Tasks*, *Tech & Engineering*, and *Games*, and 69 subdisciplines (see Figure 1) such as Robotics, Chemistry, Trading, and Agriculture, thereby fulfilling the objective of breadth in discipline coverage. The dataset includes a total of 1,559 question-answer pairs and video captions annotated and reviewed by humans. Meanwhile, for multi-faceted reasoning, MMWorld mainly contains seven kinds of questions focusing on *explanation* (explaining the phenomenon in videos), *counterfactual thinking* (answering what-if questions), *future prediction* (predicting future events), *domain expertise* (answering domain-specific inquiries), *temporal understanding* (reasoning about temporal information), and etc. Four video examples with these questions from different disciplines are depicted in Figure 1. To serve as a comprehensive benchmark, MMWorld comprises two datasets: a human-annotated dataset for evaluating MLLMs on the whole video and a synthetic dataset designed to analyze MLLMs' perception within single visual or audio modalities. We evaluate 15 MLLMs that can handle videos or image sequences on MMWorld, including both open-source (e.g., Video-LLaVA-7B (Lin et al., 2023a)) and proprietary models (GPT-4o (OpenAI, 2024) and Gemini (Team et al., 2023)).

We summarized the contributions and key findings as follows:

- We introduce MMWorld, a new benchmark designed to rigorously evaluate the capabilities of Multimodal Large Language Models (MLLMs) in world modeling through the realm of video understanding. MMWorld spans a broad spectrum of disciplines, featuring a rich array of question types for multi-faceted reasoning.

- In addition to the human-annotated dataset, we develop an automatic data collection pipeline, streamlining video content selection and question-answer generation, and construct a well-controlled synthetic dataset to analyze MLLMs within single visual or audio modalities.

---

[1]Note that the term "world model" in MMWorld is broadened from its traditional use in reinforcement learning to a more generalized sense. MMWorld is not a sufficient testbed for world model evaluation, but we believe overcoming the unique challenges presented in MMWorld is essential and necessary towards comprehensive world modeling.

Table 1: Comparison between MMWorld and previous benchmarks for real-world video understanding on a variety of criteria. Multi-faceted include Explanation (`Explain.`), Counterfactual Thinking (`Counter.`), Future Prediction (`Future.`) and Domain Expertise (`Domain.`) MMWorld is the first multi-discipline and multitask video understanding benchmark that covers wider reasoning questions, and also included first-party data annotations.

| Benchmarks | Multi-Discipline | Multi-Task | Multi-Faceted Reasoning | | | | First-Party Annotation |
|---|---|---|---|---|---|---|---|
| | | | Explain. | Counter. | Future. | Domain. | |
| MovieQA (Tapaswi et al., 2016) | | | ✓ | | | | ✓ |
| TVQA (Lei et al., 2018) | | | ✓ | | | | ✓ |
| ActivityNet-QA (Yu et al., 2019b) | | | | | | | ✓ |
| MSVD-QA (Xu et al., 2017) (Xu et al., 2016) | | | | | | | ✓ |
| MSRVTT-QA (Xu et al., 2016) | | | | | | | ✓ |
| Sports-QA (Li et al., 2024) | | | | ✓ | | ✓ | ✓ |
| VaTeX (Wang et al., 2019) | | ✓ | | | | | ✓ |
| VALUE (Li et al., 2021) | | ✓ | | | | | |
| Video-Bench (Ning et al., 2023a) | | ✓ | | | ✓ | ✓ | |
| MVBench (Li et al., 2023d) | | ✓ | | ✓ | ✓ | | |
| Perception Test (Pătrăucean et al., 2023) | | ✓ | ✓ | ✓ | ✓ | | |
| VideoMME (Fu et al., 2024) | | ✓ | | | ✓ | ✓ | ✓ |
| MMBench-Video (Fang et al., 2024) | | | | ✓ | ✓ | ✓ | ✓ |
| TempCompass (Liu et al., 2024c) | ✓ | | | | ✓ | ✓ | ✓ |
| ViLMA (Kesen et al., 2023) | ✓ | | | | ✓ | ✓ | ✓ |
| VITATECS (Li et al., 2023e) | | | | ✓ | ✓ | ✓ | ✓ |
| NExT-QA (Xiao et al., 2021) | ✓ | | ✓ | | ✓ | | ✓ |
| CVRR (Khattak et al., 2024) | | | ✓ | | ✓ | | ✓ |
| Causal-VidQA (Li et al., 2022) | | | ✓ | ✓ | ✓ | | ✓ |
| MMWorld (Ours) | ✓ | ✓ | ✓ | ✓ | ✓ | ✓ | ✓ |

- We observe that existing MLLMs still face substantial challenges posed by MMWorld. Even the best performer, GPT-4o, can only achieve a 62.54% overall accuracy, and four MLLMs particularly trained on videos perform worse than random chance.

- Although there is stll a clear gap between open-source and proprietary models, the open-source model Video-LLaVA-7B achieves the best on Embodied Tasks. It outperforms GPT-4V and Gemini Pro on Embodied Tasks by a large margin and performs similarly on Art & Sports, where spatiotemporal dynamics play a more crucial role in video understanding. This is further validated with its leading results on Temporal Understanding question type.

- In our study comparing MLLMs with average humans (non-experts), we notice some correlation between question difficulties as perceived by humans and MLLMs. However, MLLMs present different skill sets than humans in that they can answer reasonable amount of difficult questions that humans completely fail but also struggle at easy questions that humans excel at. This indicates different perception, cognition, and reasoning abilities between MLLMs and humans.

## 2 RELATED WORK

### 2.1 MULTIMODAL LARGE LANGUAGE MODELS (MLLMS)

**Emerging MLLMs** Recent advancements in Large Language Models (LLMs) (OpenAI, 2023a; Google, 2023; Touvron et al., 2023a; Chiang et al., 2023; Touvron et al., 2023b; Bai et al., 2023a) have paved the way for several multimodal counterparts in the vision-and-language domain (Dai et al., 2023; Liu et al., 2023b;a; Li et al., 2023a; Zhu et al., 2023b; Zheng et al., 2023; Bai et al., 2023b), and recently released GPT-4V (OpenAI, 2023b), followed by Gemini Vision family (Team et al., 2023). As LLMs have been applied to world modeling and simulation (Wang et al., 2024a), MLLMs now extend their capabilities beyond text and image inputs. Pretrained on large-scale, diverse datasets, these models are equipped with commonsense, domain-specific knowledge, and broad generalizability.

VideoChat (Li et al., 2023c) leverages the QFormer (Li et al., 2023b) to map visual representations to LLM (Chiang et al., 2023), and performs a multi-stage training pipeline. Otter (Li et al., 2023a) proposes to conduct instruction finetuning based on Openflamingo (Awadalla et al., 2023). PandaGPT (Su et al., 2023) employs the ImageBind (Han et al., 2023) as the backbone and finetunes it. The mPLUG-Owl (Ye et al., 2023) introduces an abstractor module to perform visual and language alignment. VideoLLaMA (Zhang et al., 2023a) introduces a frame embedding layer and

also leverages ImageBind to inject temporal and audio information into the LLM backend. Chat-UniVi (Jin et al., 2023) uses clustering to do feature fusion. LWM (Liu et al., 2024b) collects a large video and language dataset from public books and video datasets and trains a world model that is capable of processing more than millions of tokens.

These MLLMs demonstrate emerging abilities in multi-disciplinary world knowledge and excel at multi-faceted reasoning tasks, such as inverse dynamic prediction—predicting intermediate steps between previous and next states, a crucial auxiliary task for next-state prediction (Devlin, 2018; Lu et al., 2019; Paster et al., 2020) in real-world scenarios. In response to the emerging capabilities of MLLMs, we propose MMWorld to evaluate their ability to understand real-world dynamics, underlying principles, and causalities, with the ultimate goal of achieving world modeling.

**Benchmarking MLLMs** To evaluate MLLMs, there is a flourishing of analysis (Liu et al., 2024a; Zhang et al., 2023b; Jiang et al., 2022; Lu et al., 2024; Fan et al., 2024; Cui et al., 2023; Guan et al., 2024; Yu et al., 2023; Fu et al., 2023a) and the establishment of innovative benchmarks such as VisIB-Bench (Bitton et al., 2023) which evaluates models with real-world instruction-following ability given image inputs, MMMU (Yue et al., 2023) designed to access models on college-level image-question pairs that span among different disciplines, and VIM (Lu et al., 2023) which challenges the model's visual instruction following capability.

However, these recent analyses and benchmarks only cover the image input. Recently, video benchmarks such as Perception Test (Pătrăucean et al., 2023) is proposed to focus on perception and skills like memory and abstraction. However, it uses scenarios with a few objects manipulated by a person, which limits the variety of contexts. In contrast, MMWorld operates in an open-domain scenario with diverse scenes; MVBench (Li et al., 2023d), TempCompass (Liu et al., 2024c) centers on temporal understanding, while MMWorld not only includes temporal reasoning but also evaluates other multi-faceted reasoning abilities such as counterfactual thinking and domain-specific expertise; EgoSchema Mangalam et al. (2023) focuses on natural human activity and behavior, but it does not cover the broad range of disciplines that MMWorld does. MLLMs that can perfectly solve MMWorld would unlock the ability to perform multifaceted, multidisciplinary reasoning and the potential to serve as a world model.

## 2.2 Video Understanding Benchmarks

Previous video benchmarks, as shown in Table 1, focus on video understanding tasks, including activity-focused on web videos (Yu et al., 2019a), description-based question answering (Zeng et al., 2017), video completion (Fu et al., 2023b), and video infilling (Himakunthala et al., 2023). Recently, Video-Bench (Ning et al., 2023b) introduces a benchmark by collecting videos and annotations from multiple existing datasets. Mementos (Wang et al., 2024b) builds a benchmark for MLLM reasoning for input image sequences. STAR (Wu et al., 2021) builds a benchmark for situated reasoning in real-world videos. CLEVER (Yi et al., 2020) builds a benchmark containing videos focusing on objects with simple visual appearance. None of these benchmarks match the multi-discipline coverage that MMWorld provides. MMWorld, in contrast, presents a new benchmark designed to encompass interdisciplinary coverage, task diversity, and multifaceted reasoning capabilities—including future prediction, counterfactual thinking, and more—underpinned by original human annotations and integrated domain knowledge.

## 3 The MMWorld Benchmark

The MMWorld benchmark is built on three key design principles: multi-discipline coverage, multi-faceted reasoning, and temporal reasoning. It spans various disciplines that require domain expertise and incorporates diverse reasoning skills such as explanation, counterfactual thinking, and future prediction. The benchmark consists of two parts: a human-annotated dataset and a synthetic dataset. **The human-annotated dataset serves as the main testbed to evaluate MLLMs from multiple perspectives.** The synthetic dataset is divided into two subsets, each designed to assess MLLMs' perception behavior based on visual and audio inputs, respectively.

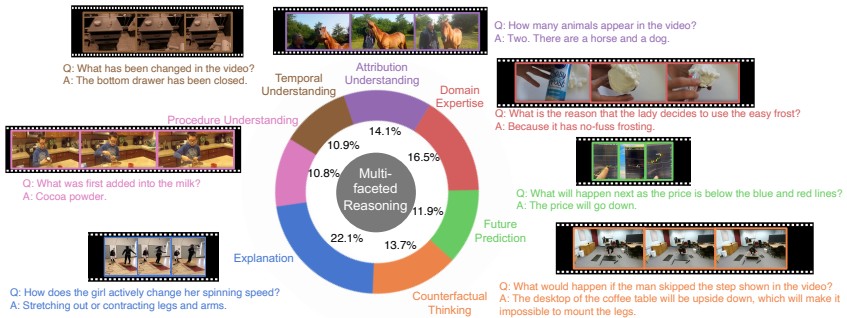

Figure 2: The questions in MMWorld are designed to evaluate seven primary understanding and reasoning abilities of models. Each question is annotated with all relevant categories. The figure showcases one example question for each reasoning category, based on its main category.

## 3.1 MANUAL DATA COLLECTION

We collect videos from YouTube with the Creative Licence in seven disciplines: Art & Sports (18.5%), Business (12.0%), Science (20.4%), Health & Medicine (12.0%), Embodied Tasks (12.0%), Tech & Engineering (12.9%), and Game (12.2%). For Art & Sports, 29 videos are collected from the SportsQA dataset (Li et al., 2024). And for Embodied Tasks, 24 videos are sourced from IKEA Assembly (Ben-Shabat et al., 2021), RT-1 (Brohan et al., 2022), and Ego4D (Grauman et al., 2022) datasets to increase video diversity.

Our manual benchmark collection takes two stages. In the first stage, we conduct a detailed examination of each of the seven primary disciplines to identify a comprehensive range of subdisciplines for inclusion in our benchmark. Our selection of videos is driven by three key principles:

1. The **first principle**, **multi-discipline** coverage, emphasizes the requirement for domain knowledge—selecting videos that inherently demand an understanding of specialized content across various disciplines; 2. The **second principle**, **multi-faceted** annotation, involves collecting videos that enable the creation of question-answer pairs from multiple perspectives to evaluate world model properties comprehensively; 3. The **third principle**, **temporal information**, prioritizes the inclusion of videos that provide meaningful content over time, as understanding temporal information is crucial for grasping world dynamics. This allows models to engage in temporal reasoning and answering questions in MMWorld requires implicit temporal reasoning, e.g., the model needs to understand temporal information to explain "why does the robot need to do the step shown in the video". We also design a "temporal understanding" question type to explicitly test models' ability to reason about temporal information (more examples can be found in Section F in the Appendix).

During the second stage, our team began the task of annotating questions, answers, and options. All annotators were asked to carefully watch the collected videos and create questions with corresponding answers and options, ensuring that understanding the video content and applying temporal reasoning were necessary to determine the correct answers. We also ensured that the clarity, correctness, and grammatical accuracy of the questions and answers were verified using GPT-4o, and that the questions could not be correctly answered without video input. We craft questions that primarily test seven aspects of multimodal video understanding also from the perspective of **multi-faceted reasoning**: 1) Explanation: Questions ask the model to elucidate the underlying logic or purpose within the video; 2) Counterfactual Thinking: Tests the model's ability to hypothesize and consider alternative outcomes; 3) Future Prediction: Aims to predict future events based on the current scenario, challenging the model's foresight; 4) Domain Expertise: Evaluates the model's depth of knowledge in specific fields, such as how to assemble a coffee table; 5) Temporal Understanding: Assesses the model's capability to reason about temporal sequences and dynamics; 6) Attribution Understanding: These questions focus on identifying cause-and-effect relationships within the video, including tasks like counting; 7) Procedure Understanding: Tests the model's ability to comprehend and explain procedural tasks shown in the video. The detailed distribution and examples are shown in Figure 2. For quality control, we ensure each annotation is cross-checked by at least two professional researchers to ensure accuracy and prevent annotation errors.

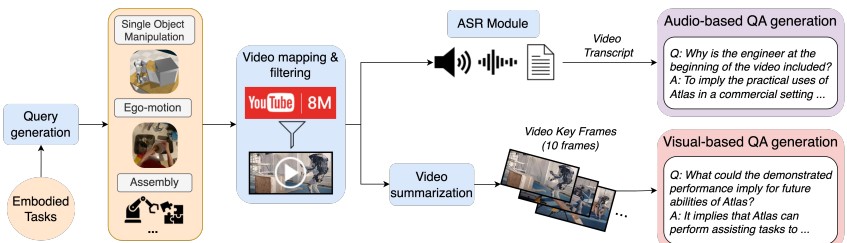

Figure 3: Schematic diagram of the synthetic data generation pipeline in MMWorld. It starts with generating subdiscipline-specific queries, followed by video retrieval from YouTube-8M (Abu-El-Haija et al., 2016) and YouTube. Keyframes are extracted for visual-based QA generation, and videos are transcribed using an ASR module for audio-based QA generation.

## 3.2 AUTOMATED DATA COLLECTION

Understanding real-world dynamics requires models to process both audio and visual modalities. To evaluate MLLMs' perception abilities in these modalities, we designed an automated data collection pipeline. This pipeline collects targeted videos and generates QA pairs based on either audio or visual information, ensuring the model's capabilities are assessed independently for each modality. By using information from a single modality to generate QA pairs, our pipeline ensures that the synthetic data remains unbiased regarding input modality.

The synthetic data generation pipeline is illustrated in Figure 3. We employ a systematic approach to gather videos with Creative Commons licenses from YouTube and the extensive YouTube-8M dataset (Abu-El-Haija et al., 2016). This method ensures a diverse and comprehensive collection of video data, which is important for the robust evaluation of multimodal video understanding models.

**Video Collection and Processing** We start with the video *Query Generator*. We start with the same seven disciplines as the manually collected dataset. For each discipline, a set of subdisciplines is defined to encapsulate a wide spectrum of topics, ensuring a diverse and comprehensive dataset. Once the queries are generated, the *Video Mapping and Filtering* step is initiated. We perform mapping of videos to YouTube-8M and online videos, constrained by a strict time limit of two minutes per query, keeping only the most pertinent videos that satisfy the predefined criteria. Simultaneously, the works in conjunction with the video transcripts to extract key terms and concepts. This iterative process refines the search parameters and enhances the semantic richness of the dataset by identifying and encoding the salient themes present in the videos. The *Video Summarization* module utilizes Query-focused video summarization techniques based on Katna[2] and UniVTG (Lin et al., 2023b). This module selects ten representative frames from each video, distilling the essence of the content while preserving the narrative context. This summarization facilitates efficient storage and quicker processing times, which are crucial for large-scale analysis.

**QA Generation** The final stage in our pipeline is the *QA / Caption Generation* module, where we leverage the capabilities of GPT-4V to generate accurate and contextually relevant questions and answers, as well as captions, based on the video frames and transcripts. This step not only provides rich annotations for each video but also equips the dataset with a multimodal dimension that supports various downstream tasks such as video QA, captioning, and more.

**Quality of the Synthetic Dataset** Human evaluators were engaged to ascertain the reasonableness of automatically generated questions and answers, ensuring that the synthetic dataset maintains a high standard of quality and relevance. The findings from this human evaluation phase are detailed in Section D of the Appendix, offering insights into the dataset's efficacy and the realism of its constructed queries and responses.

Finally, the statistics of automated curated data, which is used for the ablation study, are shown in Table 2. The taxonomy of our dataset is shown in Figure 1. We note that only a portion of the subdisciplines are shown due to space concerns. Please refer to the Appendix for full information.

---

[2]https://github.com/keplerlab/katna

Table 2: Key Statistics of the MMWorld Benchmark. The main subset is the human-annotated subset. Synthetic Subset I contains generated QA pairs focused exclusively on the audio content, while Synthetic Subset II contains QA pairs focused exclusively on the visual content of the video.

| Statistics | Main Subset | Synthetic I | Synthetic II |
|---|---|---|---|
| #Discipline/#Subdiscipline | 7/61 | 7/51 | 7/54 |
| #Videos | 417 | 746 | 747 |
| #QA pairs | 1,559 | 2,969 | 2,099 |
| Avg Video Lengths (s) | 102.3 | 103.4 | 115.8 |
| Avg #Questions per Video | 4.05 | 3.98 | 2.81 |
| Avg #Options | 3.90 | 4.00 | 4.00 |
| Avg Question Length | 11.39 | 15.12 | 17.56 |
| Avg Option Length | 7.27 | 6.01 | 5.19 |
| Avg Answer Length | 6.42 | 6.71 | 5.67 |
| Avg Caption Length | 27.00 | 71.87 | 82.33 |
| # Unique Words in Questions | 1,913 | 2,528 | 2,279 |
| # Unique Words in Answers | 2,292 | 2,981 | 2,657 |

## 4 EXPERIMENTS

### 4.1 EXPERIMENTAL SETTINGS

In our study, we compare MLLM's performance on the MMWorld benchmark, including GPT-4o (OpenAI, 2024), GPT-4V (OpenAI, 2023b), Gemini Pro (Team et al., 2023), Claude-3.5-Sonnet Anthropic (2024), Video-Chat (Li et al., 2023c), Video-ChatGPT (Maaz et al., 2024), Video-LLaMA (Zhang et al., 2023a), Video-LLaVA (Lin et al., 2023a), ChatUnivi (Jin et al., 2023), mPLUG-Owl (Ye et al., 2023), Otter (Li et al., 2023a), ImageBind-LLM (Han et al., 2023), PandaGPT (Su et al., 2023), LWM (Liu et al., 2024b), and X-Instruct-BLIP (Panagopoulou et al., 2023). For proprietary model, we adhere to the default settings provided by their official APIs. They both take ten image frames extracted from the video content as the input. The Gemini Pro is set to process visual input and configured with safety settings to filter a range of harmful content. The configuration thresholds are set to 'BLOCK_NONE'. For PandaGPT, we set 'top_p' to 0.7 and 'temperature' to 0.5. For VideoChat, we set 'max_frames' to 100. For X-Instruct-BLIP, the model is implemented using four image frames. We use GPT-4-32K as the judge for judging whether the model answer is correct when it can not mapped to the option letter using the rule-based method. For others, we all use the default setting. All inferences are run on a NVIDIA A6000 workstation. The detailed implementation is given in the Appendix.

### 4.2 EVALUATION STRATEGY

Our dataset contains multiple-choice questions and captions corresponding to each video, supporting tasks such as video question answering and video captioning. In our evaluation setup, we focus on video question answering by measuring a model's accuracy in selecting the correct answer from the provided options. This method is straightforward to quantify and provides objective assessment. However, one challenge is reliably mapping the model's predictions to one of the predefined choices.

To address this, we employ two mapping strategies. We employ two mapping strategies. The first method employs automated scripts to parse the models' predictions and compare the parsed results with the ground truth, similar to the approach used in (Yue et al., 2023); The second method involves models freely generating answers, which are then evaluated by GPT-4. Given the question, correct answer, and model's prediction, GPT-4 returns a True or False judgment. This approach is based on recent works in model evaluation (Maaz et al., 2024; Hsu et al., 2023; Hackl et al., 2023; Liu et al., 2023c).

We validated the second GPT-4-based evaluation approach with human evaluators, showing an error rate of only 4.76% across 189 examples, demonstrating its reliability as an evaluator. Detailed results for human evaluation and both evaluation strategies are provided in Appendix. All results presented in the main paper are based on the second evaluation approach.

Table 3: MLLM accuracy across diverse disciplines (averaging over three runs). GPT-4V and Gemini Pro lead at most disciplines and achieve the best overall accuracy. The best open-source model Video-LLaVA-7B outperforms them on Embodied Tasks and perform similarly on Art & Sports. All data are annotated by humans.

| Model | Art& Sports | Business | Science | Health& Medicine | Embodied Tasks | Tech& Engineering | Game | Average |
|---|---|---|---|---|---|---|---|---|
| Random Choice | 25.03 | 25.09 | 26.44 | 25.00 | 26.48 | 30.92 | 25.23 | 26.31 |
| *Proprietary MLLMs* | | | | | | | | |
| GPT-4o (OpenAI, 2024) | $47.87_{\pm1.47}$ | $\mathbf{91.14}_{\pm0.87}$ | $\mathbf{73.78}_{\pm2.88}$ | $\mathbf{83.33}_{\pm1.47}$ | $62.94_{\pm3.47}$ | $\mathbf{75.53}_{\pm2.61}$ | $\mathbf{80.32}_{\pm2.05}$ | $\mathbf{62.54}_{\pm0.79}$ |
| Claude-3.5-Sonnet (Anthropic, 2024) | $\mathbf{54.58}_{\pm0.45}$ | $63.87_{\pm0.40}$ | $59.85_{\pm1.28}$ | $54.51_{\pm1.28}$ | $30.99_{\pm0.40}$ | $58.87_{\pm0.61}$ | $59.44_{\pm0.68}$ | $54.54_{\pm0.29}$ |
| GPT-4V (OpenAI, 2023b) | $36.17_{\pm0.58}$ | $81.59_{\pm1.74}$ | $66.52_{\pm1.86}$ | $73.61_{\pm0.49}$ | $55.48_{\pm2.70}$ | $61.35_{\pm1.00}$ | $73.49_{\pm1.97}$ | $52.30_{\pm0.49}$ |
| Gemini Pro (Team et al., 2023) | $37.12_{\pm2.68}$ | $76.69_{\pm2.16}$ | $62.81_{\pm1.83}$ | $76.74_{\pm1.30}$ | $43.59_{\pm0.33}$ | $69.86_{\pm2.01}$ | $66.27_{\pm2.60}$ | $51.02_{\pm1.35}$ |
| *Open-source MLLMs* | | | | | | | | |
| Video-LLaVA-7B (Lin et al., 2023a) | $35.91_{\pm0.96}$ | $51.28_{\pm0.87}$ | $56.30_{\pm0.76}$ | $32.64_{\pm0.49}$ | $\mathbf{63.17}_{\pm1.44}$ | $58.16_{\pm1.00}$ | $49.00_{\pm3.16}$ | $44.60_{\pm0.58}$ |
| Video-Chat-7B (Li et al., 2023c) | $39.53_{\pm0.06}$ | $51.05_{\pm0.00}$ | $30.81_{\pm0.21}$ | $46.18_{\pm0.49}$ | $40.56_{\pm0.57}$ | $39.36_{\pm0.00}$ | $44.98_{\pm0.57}$ | $40.11_{\pm0.06}$ |
| ChatUnivi-7B (Jin et al., 2023) | $24.47_{\pm0.49}$ | $60.84_{\pm1.51}$ | $52.00_{\pm0.73}$ | $61.11_{\pm1.96}$ | $46.15_{\pm2.06}$ | $56.74_{\pm1.33}$ | $52.61_{\pm2.84}$ | $39.47_{\pm0.42}$ |
| mPLUG-Owl-7B (Ye et al., 2023) | $29.16_{\pm1.62}$ | $64.10_{\pm1.84}$ | $47.41_{\pm3.29}$ | $60.07_{\pm1.30}$ | $23.78_{\pm2.47}$ | $41.84_{\pm5.09}$ | $62.25_{\pm3.16}$ | $38.94_{\pm1.52}$ |
| Video-ChatGPT-7B (Maaz et al., 2024) | $26.84_{\pm0.69}$ | $39.16_{\pm3.02}$ | $36.45_{\pm1.31}$ | $53.12_{\pm0.00}$ | $36.60_{\pm3.25}$ | $41.49_{\pm1.74}$ | $36.55_{\pm2.27}$ | $33.27_{\pm0.97}$ |
| PandaGPT-7B (Su et al., 2023) | $25.33_{\pm0.54}$ | $42.66_{\pm3.02}$ | $39.41_{\pm2.67}$ | $38.54_{\pm3.07}$ | $35.43_{\pm0.87}$ | $41.84_{\pm2.79}$ | $40.16_{\pm4.65}$ | $32.48_{\pm0.45}$ |
| ImageBind-LLM-7B (Han et al., 2023) | $24.82_{\pm0.16}$ | $42.66_{\pm0.99}$ | $32.15_{\pm1.11}$ | $30.21_{\pm1.47}$ | $46.85_{\pm1.14}$ | $41.49_{\pm1.50}$ | $41.37_{\pm0.57}$ | $31.75_{\pm0.14}$ |
| X-Instruct-BLIP-7B (Panagopoulou et al., 2023) | $21.08_{\pm0.27}$ | $15.85_{\pm0.87}$ | $22.52_{\pm1.11}$ | $28.47_{\pm0.49}$ | $18.41_{\pm1.44}$ | $22.34_{\pm0.87}$ | $26.10_{\pm0.57}$ | $21.36_{\pm0.18}$ |
| LWM-1M-JAX (Liu et al., 2024b) | $12.04_{\pm0.53}$ | $17.48_{\pm0.57}$ | $15.41_{\pm0.91}$ | $20.49_{\pm0.98}$ | $25.87_{\pm1.98}$ | $21.99_{\pm2.19}$ | $11.65_{\pm3.01}$ | $15.39_{\pm0.32}$ |
| Otter-7B (Li et al., 2023a) | $17.12_{\pm1.17}$ | $18.65_{\pm0.87}$ | $9.33_{\pm0.36}$ | $6.94_{\pm0.98}$ | $13.29_{\pm1.51}$ | $15.96_{\pm1.74}$ | $15.26_{\pm0.57}$ | $14.99_{\pm0.77}$ |
| Video-LLaMA-2-13B (Zhang et al., 2023a) | $6.15_{\pm0.44}$ | $21.21_{\pm0.66}$ | $22.22_{\pm1.45}$ | $31.25_{\pm1.70}$ | $15.38_{\pm1.14}$ | $19.15_{\pm1.74}$ | $24.90_{\pm5.93}$ | $14.03_{\pm0.29}$ |

Table 4: Results of different MLLMs on multi-faceted reasoning. All data are annotated by humans.

| Model | Explanation | Counterfactual Thinking | Future Prediction | Domain Expertise | Attribution Understanding | Temporal Understanding |
|---|---|---|---|---|---|---|
| *Proprietary MLLMs* | | | | | | |
| GPT-4o (OpenAI, 2024) | $\mathbf{56.68}_{\pm0.72}$ | $\mathbf{75.88}_{\pm1.47}$ | $\mathbf{82.48}_{\pm0.69}$ | $\mathbf{69.05}_{\pm0.49}$ | $\mathbf{65.10}_{\pm1.15}$ | $\mathbf{40.90}_{\pm2.42}$ |
| GPT-4V (OpenAI, 2023b) | $44.90_{\pm0.07}$ | $64.90_{\pm0.58}$ | $78.59_{\pm1.55}$ | $61.07_{\pm0.17}$ | $59.61_{\pm0.85}$ | $27.17_{\pm1.00}$ |
| Claude-3.5-Sonnet (Anthropic, 2024) | $51.94_{\pm0.23}$ | $62.75_{\pm0.16}$ | $71.78_{\pm0.40}$ | $66.79_{\pm0.45}$ | $40.00_{\pm0.55}$ | $25.77_{\pm0.46}$ |
| Gemini Pro (Team et al., 2023) | $48.58_{\pm1.07}$ | $65.49_{\pm0.42}$ | $65.45_{\pm1.05}$ | $53.87_{\pm1.31}$ | $43.92_{\pm1.40}$ | $24.65_{\pm1.00}$ |
| *Open-source MLLMs* | | | | | | |
| Video-LLaVA (Lin et al., 2023a) | $42.46_{\pm0.61}$ | $42.55_{\pm0.85}$ | $64.96_{\pm0.69}$ | $47.86_{\pm0.58}$ | $36.86_{\pm1.95}$ | $34.45_{\pm1.19}$ |
| Video-Chat-7B (Li et al., 2023c) | $41.66_{\pm0.06}$ | $43.73_{\pm0.32}$ | $45.74_{\pm0.20}$ | $40.95_{\pm0.10}$ | $30.59_{\pm0.00}$ | $25.77_{\pm0.23}$ |
| Video-ChatGPT-7B (Maaz et al., 2024) | $32.13_{\pm0.38}$ | $39.02_{\pm1.12}$ | $47.45_{\pm2.09}$ | $33.69_{\pm1.08}$ | $21.18_{\pm2.00}$ | $23.53_{\pm0.76}$ |
| ImageBind-LLM-7B (Han et al., 2023) | $29.51_{\pm0.27}$ | $26.86_{\pm0.58}$ | $50.61_{\pm0.20}$ | $33.93_{\pm0.17}$ | $34.90_{\pm1.40}$ | $19.89_{\pm0.91}$ |
| PandaGPT-7B (Su et al., 2023) | $29.55_{\pm0.41}$ | $37.45_{\pm1.80}$ | $46.47_{\pm1.05}$ | $33.93_{\pm0.45}$ | $26.27_{\pm2.24}$ | $28.01_{\pm0.82}$ |
| ChatUnivi-7B (Jin et al., 2023) | $33.91_{\pm0.31}$ | $48.82_{\pm0.48}$ | $61.80_{\pm0.53}$ | $45.95_{\pm0.68}$ | $33.33_{\pm0.64}$ | $22.97_{\pm0.91}$ |
| Video-LLaMA-2-13B (Zhang et al., 2023a) | $10.55_{\pm0.29}$ | $23.92_{\pm0.97}$ | $25.30_{\pm1.11}$ | $16.31_{\pm1.03}$ | $8.63_{\pm0.85}$ | $6.16_{\pm1.00}$ |
| X-Instruct-BLIP-7B (Panagopoulou et al., 2023) | $23.05_{\pm0.24}$ | $15.29_{\pm0.28}$ | $27.25_{\pm0.53}$ | $21.07_{\pm0.51}$ | $24.31_{\pm0.64}$ | $11.20_{\pm0.82}$ |
| LWM-1M-JAX (Liu et al., 2024b) | $11.62_{\pm0.39}$ | $18.82_{\pm0.55}$ | $30.66_{\pm0.34}$ | $17.98_{\pm0.26}$ | $21.57_{\pm0.85}$ | $7.00_{\pm0.46}$ |
| Otter-7B (Li et al., 2023a) | $16.91_{\pm0.54}$ | $10.98_{\pm0.42}$ | $15.82_{\pm0.20}$ | $13.10_{\pm0.68}$ | $17.65_{\pm0.00}$ | $9.52_{\pm1.00}$ |
| mPLUG-Owl-7B (Ye et al., 2023) | $35.20_{\pm1.17}$ | $49.61_{\pm1.31}$ | $55.47_{\pm1.58}$ | $47.74_{\pm1.07}$ | $24.71_{\pm2.00}$ | $20.17_{\pm0.69}$ |

## 4.3 MAIN EVALUATION RESULTS ON HUMAN-ANNOTATED DATA

We show in Table 3 the main evaluation results of different MLLMs. Among these, GPT-4o emerges as the top performer, followed by Claude-3.5-Sonnet. Video-LLaVA also demonstrates strong results, primarily due to the extensive training data which consists of 558K LAION-CCSBU image-text pairs and 702K video-text pairs from WebVid (Bain et al., 2021). Its superior performance may also be attributed to the adoption of CLIP ViT-L/14 trained in LanguageBind (Lin et al., 2023a) as its vision model and the inclusion of a large volume of image-video-text pairings within the training data. On the other hand, models like Otter and LWM perform poorly across most disciplines, possibly due to their weaker backbone and architecture used. Otter uses the LLaMA-7B language encoder and a CLIP ViT-L/14 vision encoder, both of which are frozen, with only the Perceiver resampler (Awadalla et al., 2023) module fine-tuned, which may lead to the lower performance. Additionally, four MLLMs perform even worse than random, highlighting the challenging nature of MMWorld.

**Study on Multi-faceted Reasoning** Table 4 illustrates the multi-faceted reasoning performance of each MLLM. GPT-4o emerges as the strongest model across all facets. Notably, in temporal understanding, the open-sourced Video-LLaVA outperforms all other models except GPT-4o, likely due to its extensive training on high temporal resolution video data, enhancing its spatio-temporal reasoning abilities. This is further reflected in its high scores on Embodied Tasks (the best) and Art & Sports, both of which involve dense spatio-temporal information, as shown in Table 3.

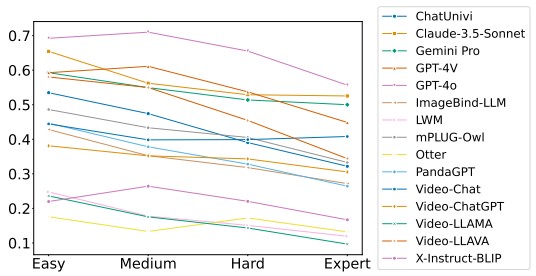 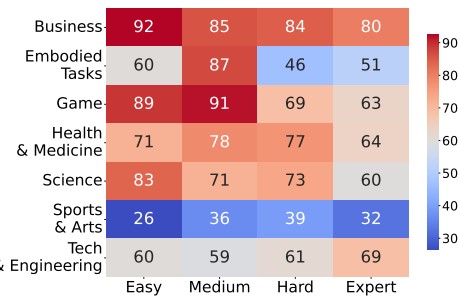

(a) Accuracy of MLLMs at difficulty levels for average humans.

(b) GPT-4V results by disciplines at different difficulty levels for average humans.

Figure 4: Model performance at different difficulty levels for average humans. Average human difficulty levels are defined by 3 turkers' performance per question: Easy (3/3 correct answers), medium (2/3 correct), hard (1/3 correct), and expert (0/3 correct).

**Study on MLLM Performance at Different Difficulty Levels for Average Humans**

Figure 4a indicate some correlation between the difficulty levels as perceived by humans and the performance of MLLMs. The difficulty levels are defined based on the **average human performance**. MLLMs generally follow a trend where accuracy decreases as the difficulty level increases, which aligns with human performance patterns. However, the correlation is not perfect, suggesting that while models and humans share some common ground in understanding question difficulty, there are also notable differences in their capabilities. The data reveals that MLLMs exhibit different skill sets compared to humans. As highlighted in Figure 4b, models like GPT-4V can correctly answer expert-level questions that humans often get wrong, particularly in disciplines such as Business and Health & Medicine, where humans often struggle, yet they sometimes falter on easier questions, likely due to the lack of contextual understanding. Notably, discrepancies in disciplines like Art & Sports and Tech & Engineering highlight areas where MLLMs' performance does not align with human results, suggesting different perception, cognition, and reasoning abilities in handling abstract concepts. These differences suggest that MLLMs can complement human capabilities, offering potential for enhanced task performance by combining the data-driven insights of models with human intuition and contextual knowledge.

**Error Analysis** To gain deeper insights into the limitations of current open-sourced MLLMs and provide guidance for developing next-generation models, we prompted the models to explain their reasoning, particularly when errors occurred. We grouped and identified common error patterns into seven distinct categories. We conducted a comparative test by posing the error-inducing questions for GPT-4V to other MLLMs, as GPT-4V was used as a representative model due to its strong performance and its ability to highlight errors common across MLLMs.

Our analysis revealed that Video-LLaVA exhibited the lowest error frequencies among open-source MLLMs Its superior performance, particularly in reducing Visual Perception Errors (PE), Hallucination Errors (HE), and Reasoning Errors (RE), can also be linked to its use of the CLIP ViT-L/14 model in LanguageBind (Zhu et al., 2023a). In contrast, mPLUG-Owl showed higher rates of Visual Perception Errors, possibly due to its reliance on weaker video embedder architectures. Furthermore, VideoChat outperformed Video-LLaMA due to its GMHRA (Li et al., 2023c) module for temporal aggregation, demonstrating the importance of effective temporal aggregation in reducing errors. Common trends across all models included frequent hallucination errors and a lack of domain-specific knowledge, highlighting the need for accurate, noise-free training data and suggesting that techniques like Reinforcement Learning from Human Feedback (RLHF) (Ouyang et al., 2022) could help mitigate these issues. While current MLLMs demonstrate strong multi-disciplinary world knowledge, they could benefit from enhanced domain-specific expertise, potentially through retrieval-based methods. Detailed qualitative examples and further analysis are provided in the Appendix.

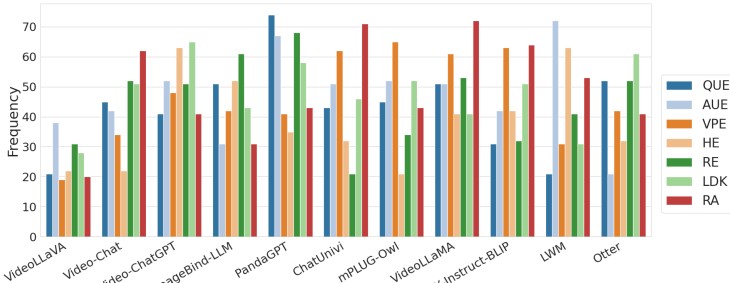

Figure 5: The frequency of different error types across various MLLMs. For each error type, 100 examples were evaluated. Error types are abbreviated as follows: QUE (Question Understanding Error), AUE (Audio Understanding Error), VPE (Visual Perception Error), HE (Hallucination Error), RE (Reasoning Error), LDK (Lack of Domain Knowledge), and RA (Reject to Answer).

Table 5: Performance on Synthetic Subset I (Audio) and II (Visual). Synthetic Subset I contains QAs based solely on the audio content, while Synthetic Subset II focuses exclusively on the visual content of the video. We evaluated four MLLMs processing both audio and visual inputs along with Gemini Pro (for the audio setting, only providing the question).

| Model | Art&Sports | | Business | | Science | | Health&Medicine | | Embodied Tasks | | Tech&Engineering | | Game | | Average | |
|---|---|---|---|---|---|---|---|---|---|---|---|---|---|---|---|---|
| | Audio | Visual | Audio | Visual | Audio | Visual | Audio | Visual | Audio | Visual | Audio | Visual | Audio | Visual | Audio | Visual |
| Random Choice | 31.59 | 30.14 | 31.18 | 26.58 | 36.98 | 32.89 | 38.74 | 32.64 | 32.81 | 31.25 | 27.23 | 32.60 | 32.01 | 30.78 | 32.44 | 30.91 |
| Video-Chat (Li et al., 2023c) | **33.98** | 32.48 | **46.47** | 41.46 | **41.86** | 39.15 | **45.95** | 36.81 | 32.81 | 46.88 | **37.48** | 35.91 | **32.98** | 46.70 | **38.82** | 39.07 |
| ChatUnivi (Jin et al., 2023) | 30.03 | 43.22 | 30.19 | 52.85 | 38.75 | 54.59 | 34.76 | 50.69 | 20.14 | 40.63 | 24.17 | 46.41 | 29.98 | 45.44 | 31.82 | 48.44 |
| Video-LLaMA (Zhang et al., 2023a) | 30.15 | 30.23 | 36.18 | 33.17 | 31.33 | 31.34 | 30.90 | 32.78 | **33.13** | 30.05 | 31.18 | 30.55 | 20.49 | 27.20 | 29.08 | 30.47 |
| Otter (Li et al., 2023a) | 14.22 | 16.82 | 16.77 | 14.24 | 16.12 | 17.00 | 19.82 | 13.19 | 10.94 | 12.50 | 15.63 | 12.43 | 6.65 | 10.44 | 12.83 | 13.41 |
| Gemini Pro (Team et al., 2023) | 20.88 | **61.38** | 29.43 | **77.35** | 30.62 | **74.26** | 30.14 | **81.53** | 22.57 | **70.31** | 18.83 | **66.22** | 29.96 | **65.01** | 24.45 | **69.97** |

## 4.4 STUDY ON MODALITY OF PERCEPTION ON SYNTHETIC DATA

We conducted ablation studies to evaluate how well MLLMs can perceive the world when limited to a single modality (audio or visual) using the synthetic dataset of MMWorld. In these experiments, we isolated scenarios where only one modality—either audio or visual—was available. Table 5 presents the results, which assess the models' ability to interpret spoken language, background noises, and other audio elements without visual context, as well as their visual perception without any audio input. For the visual perception test, Gemini Pro performed the best, demonstrating its strong ability to process visual information. Interestingly, Video-Chat exhibited better audio perception than ChatUnivi, despite its poorer visual perception. This may be attributed to its use of the Whisper (Radford et al., 2022) speech recognition model. It also explains that in Table 3, Video-Chat outperforms ChatUnivi in the Art & Sports discipline, which requires a greater understanding of music, voice, and background audio. However, in other disciplines such as Science and Health & Medicine, Video-Chat's performance is significantly worse.

## 5 CONCLUSION

Our MMWorld Benchmark represents a significant step forward in the quest for advanced multi-modal language models capable of understanding complex video content. By presenting a diverse array of videos across seven disciplines, accompanied by questions that challenge models to demonstrate explanation, counterfactual thinking, future prediction, and domain expertise, we have created a rigorous testing ground for the next generation of AI. While using LLMs for data generation can introduce hallucination issues, these challenges are manageable and are commonly addressed (Wang et al., 2024c; Shen et al., 2023). Another potential risk is the misuse of MLLMs for surveillance or privacy invasion. The ability of models to understand video content and perform reasoning could be exploited to monitor individuals without their consent, leading to serious ethical and legal concerns regarding privacy.

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

**Ethics Statement**    In line with the ICLR Code of Ethics, we acknowledge our responsibility to adhere to ethical principles throughout the entirety of our research. Our work does not involve human subjects, and the datasets we used are available in the submitted supplementary material and do not raise any concerns regarding privacy or security issues. The evaluation of models in this paper focuses on publicly available multimodal larger language models, and no sensitive or personally identifiable information was involved in this process. While our work benchmarks existing multimodal large language models via multi-discipline, multi-faceted world model evaluation, we recognize the potential risks of unintended bias and fairness issues in these models, which may have inherited biases from their training data. We encourage future research to address these concerns in the development of more inclusive and fair models. There are no conflicts of interest or sponsorship influencing this research, and our work fully complies with legal and ethical standards.

**Reproducibility Statement**    To ensure the reproducibility of our work, we provide extensive details on our methodology, datasets, and evaluation setup in the main paper and the Appendix. The datasets used are available in the supplementary material, and their collection and annotation steps are described in Section 3 of the paper. We also provide detailed descriptions of the experimental setup, including hyperparameters, model architectures, in the main paper and Appendix. All evaluation protocols and metrics are explained to facilitate replication of our results.

## A    OVERVIEW OF THE APPENDIX

This Appendix is organized as follows:

- Section B contains additional experimental results;

- Section C contains the implementation details;

- Section D contains the settings and results from human evaluations;

- Section E contains the error analysis;

- Section F contains the data examples from MMWorld;

- Section G contains additional data statistics of MMWorld;

## B    ADDITIONAL RESULTS

### B.1    RESULTS ACROSS DIFFERENT SEED FOR EACH MODEL

In Table 6, we show detailed results using three different seeds for each evaluated models.

### B.2    RESULTS FROM AMAZON TURKERS

Table 7 presents the evaluation results from three sets of Amazon Turkers across various disciplines. The results indicate that there is slightly variability in performance across different human evaluators.

### B.3    RESULTS FOR THE TWO DIFFERENT EVALUATION STRATEGIES

In Table 8, we give additional evaluation results for different MLLMs evaluated in this paper. For closed-source models, the evaluation pipeline is the one used in the main paper, which involves utilizing GPT-4V as a judger. The process consists of presenting GPT-4V with the question, a corresponding answer generated by the baseline model, and the set of possible options. GPT-4V then assesses whether the model-generated answer is accurate within the given context; Another is open-ended generation where we employ a two-step methodology. We first prompt each model to do open-ended generation. Subsequently, we prompt the model to align its generative response with one of the predefined options: 'a', 'b', 'c', or 'd'.

Table 6: Detailed results (%) of model performance, measured as accuracy percentages across diverse disciplines for three runs. The random choice baseline involves shuffling candidate answers for each video question before consistently selecting answer 'a'. GPT-4V and Gemini Pro utilize 10 image frames extracted from the video content.

| Model | Art& Sports | Business | Science | Health& Medicine | Embodied Tasks | Tech& Engineering | Game | Average |
|---|---|---|---|---|---|---|---|---|
| GPT-4o-seed 1 (OpenAI, 2024) | 47.10 | 92.31 | 75.11 | 81.25 | 65.03 | 72.34 | 78.31 | 62.22 |
| GPT-4o-seed 2 (OpenAI, 2024) | 46.58 | 90.91 | 69.78 | 84.38 | 65.73 | 75.53 | 83.13 | 61.77 |
| GPT-4o-seed 3 (OpenAI, 2024) | 49.94 | 90.21 | 76.44 | 84.38 | 58.04 | 78.72 | 79.52 | 63.63 |
| Claude-3.5-seed 1 (Anthropic, 2024) | 54.32 | 64.34 | 59.11 | 53.12 | 30.77 | 59.57 | 59.04 | 54.27 |
| Claude-3.5-seed 2 (Anthropic, 2024) | 54.32 | 63.64 | 61.33 | 54.17 | 30.77 | 58.51 | 59.04 | 54.52 |
| Claude-3.5-seed 3 (Anthropic, 2024) | 55.10 | 63.64 | 59.11 | 56.25 | 31.47 | 58.51 | 60.24 | 54.84 |
| GPT-4V-seed 1 (OpenAI, 2023b) | 36.90 | 79.72 | 64.00 | 73.96 | 51.75 | 60.64 | 71.08 | 51.64 |
| GPT-4V-seed 2 (OpenAI, 2023b) | 35.48 | 83.92 | 68.44 | 73.96 | 58.04 | 60.64 | 75.90 | 52.79 |
| GPT-4V-seed 3 (OpenAI, 2023b) | 36.13 | 81.12 | 67.11 | 72.92 | 56.64 | 62.77 | 73.49 | 52.47 |
| Gemini Pro-seed 1 (Team et al., 2023) | 40.90 | 79.72 | 60.44 | 78.12 | 43.36 | 71.28 | 65.06 | 52.92 |
| Gemini Pro-seed 2 (Team et al., 2023) | 35.10 | 75.52 | 63.11 | 75.00 | 44.06 | 71.28 | 69.88 | 50.16 |
| Gemini Pro-seed 3 (Team et al., 2023) | 35.35 | 74.83 | 64.89 | 77.08 | 43.36 | 67.02 | 63.86 | 49.97 |
| Video-LLaVA-seed 1 (Lin et al., 2023a) | 34.58 | 51.05 | 57.33 | 32.29 | 61.54 | 57.45 | 50.60 | 43.94 |
| Video-LLaVA-seed 2 (Lin et al., 2023a) | 36.77 | 52.45 | 56.00 | 32.29 | 65.03 | 57.45 | 51.81 | 45.35 |
| Video-LLaVA-seed 3 (Lin et al., 2023a) | 36.39 | 50.35 | 55.56 | 33.33 | 62.94 | 59.57 | 44.58 | 44.52 |
| Video-Chat-seed 1 (Li et al., 2023c) | 39.48 | 51.05 | 30.67 | 46.88 | 39.86 | 39.36 | 44.58 | 40.03 |
| Video-Chat-seed 2 (Li et al., 2023c) | 39.48 | 51.05 | 30.67 | 45.83 | 41.26 | 39.36 | 45.78 | 40.15 |
| Video-Chat-seed 3 (Li et al., 2023c) | 39.61 | 51.05 | 31.11 | 45.83 | 40.56 | 39.36 | 44.58 | 40.15 |
| mPLUG-Owl-seed 1 (Ye et al., 2023) | 31.35 | 65.73 | 45.78 | 61.46 | 28.67 | 48.94 | 65.06 | 41.05 |
| mPLUG-Owl-seed 2 (Ye et al., 2023) | 28.65 | 65.03 | 44.44 | 58.33 | 21.68 | 37.23 | 57.83 | 37.52 |
| mPLUG-Owl-seed 3 (Ye et al., 2023) | 27.48 | 61.54 | 52.00 | 60.42 | 20.98 | 39.36 | 63.86 | 38.23 |
| ChatUnivi-seed 1 (Jin et al., 2023) | 24.13 | 60.14 | 52.00 | 62.50 | 48.95 | 56.38 | 56.63 | 39.77 |
| ChatUnivi-seed 2 (Jin et al., 2023) | 25.16 | 62.94 | 51.11 | 62.50 | 44.06 | 58.51 | 50.60 | 39.77 |
| ChatUnivi-seed 3 (Jin et al., 2023) | 24.13 | 59.44 | 52.89 | 58.33 | 45.45 | 55.32 | 50.60 | 38.87 |
| Video-ChatGPT-seed 1 (Maaz et al., 2024) | 26.71 | 34.97 | 34.67 | 53.12 | 37.76 | 41.49 | 34.94 | 32.59 |
| Video-ChatGPT-seed 2 (Li et al., 2023c) | 27.74 | 41.96 | 36.89 | 53.12 | 39.86 | 43.62 | 39.76 | 34.64 |
| Video-ChatGPT-seed 3 (Li et al., 2023c) | 27.74 | 41.96 | 36.89 | 53.12 | 39.86 | 43.62 | 39.76 | 34.64 |
| PandaGPT-seed 1 (Su et al., 2023) | 26.06 | 44.06 | 38.22 | 41.67 | 35.66 | 39.36 | 42.17 | 32.97 |
| PandaGPT-seed 2 (Su et al., 2023) | 24.77 | 45.45 | 36.89 | 34.38 | 34.27 | 40.43 | 44.58 | 31.88 |
| PandaGPT-seed 3 (Su et al., 2023) | 25.16 | 38.46 | 43.11 | 39.58 | 36.36 | 45.74 | 33.73 | 32.58 |
| ImageBind-LLM-seed 1 (Han et al., 2023) | 24.77 | 41.96 | 30.67 | 31.25 | 46.85 | 43.62 | 40.96 | 31.62 |
| ImageBind-LLM-seed 2 (Han et al., 2023) | 25.03 | 41.96 | 32.44 | 31.25 | 45.45 | 40.43 | 40.96 | 31.69 |
| ImageBind-LLM-seed 3 (Han et al., 2023) | 24.65 | 44.06 | 33.33 | 28.12 | 48.25 | 40.43 | 42.17 | 31.94 |
| X-Instruct-BLIP-seed 1 (Panagopoulou et al., 2023) | 21.42 | 14.69 | 22.22 | 29.17 | 16.78 | 21.28 | 26.51 | 21.23 |
| X-Instruct-BLIP-seed 2 (Panagopoulou et al., 2023) | 20.77 | 16.78 | 24.00 | 28.12 | 20.28 | 22.34 | 25.30 | 21.62 |
| X-Instruct-BLIP-seed 3 (Panagopoulou et al., 2023) | 21.03 | 16.08 | 21.33 | 28.12 | 18.18 | 23.40 | 26.51 | 21.23 |
| LWM-seed 1 (Liu et al., 2024b) | 11.35 | 18.18 | 16.44 | 19.79 | 24.48 | 24.47 | 10.84 | 15.20 |
| LWM-seed 2 (Liu et al., 2024b) | 12.13 | 17.48 | 15.56 | 19.79 | 24.48 | 22.34 | 8.43 | 15.14 |
| LWM-seed 3 (Liu et al., 2024b) | 12.65 | 16.78 | 14.22 | 21.88 | 28.67 | 19.15 | 15.66 | 15.84 |
| Otter-seed 1 (Li et al., 2023a) | 18.45 | 19.58 | 8.89 | 8.33 | 14.69 | 15.96 | 14.46 | 15.84 |
| Otter-seed 2 (Li et al., 2023a) | 17.29 | 17.48 | 9.33 | 6.25 | 13.99 | 18.09 | 15.66 | 15.14 |
| Otter-seed 3 (Li et al., 2023a) | 15.61 | 18.88 | 9.78 | 6.25 | 11.19 | 13.83 | 15.66 | 13.98 |
| Video-LLaMA-seed 1 (Zhang et al., 2023a) | 5.55 | 21.68 | 24.00 | 29.17 | 15.38 | 21.28 | 18.07 | 13.66 |
| Video-LLaMA-seed 2 (Zhang et al., 2023a) | 6.58 | 20.28 | 20.44 | 31.25 | 13.99 | 17.02 | 32.53 | 14.05 |
| Video-LLaMA-seed 3 (Zhang et al., 2023a) | 6.32 | 21.68 | 22.22 | 33.33 | 16.78 | 19.15 | 24.10 | 14.37 |

Table 7: Performance (%) of different set of turkers

| Model | Art& Sports | Business | Science | Health& Medicine | Embodied Tasks | Tech& Engineering | Game& | Average |
|---|---|---|---|---|---|---|---|---|
| Turker Set 1 | 25.224 | 39.860 | 32.444 | 40.625 | 51.049 | 50.000 | 40.964 | 33.227 |
| Turker Set 2 | 30.452 | 46.154 | 35.556 | 42.708 | 53.846 | 51.064 | 46.988 | 37.652 |
| Turker Set 3 | 26.710 | 41.958 | 36.889 | 46.875 | 53.147 | 42.553 | 38.554 | 34.830 |

## B.4 ADDITIONAL EVALUATION RESULTS USING OPEN-SOURCED EVALUATOR AS THE EVALUATOR

In addition to GPT-4V, we also experimented with using the open-sourced Video-LLaVA model as an evaluator. The average accuracy of various models evaluated with this method is shown in Table 10. The rankings are consistent with those obtained using GPT-4V in the main paper, highlighting the versatility of our benchmark, which supports multiple evaluator options beyond GPT-4V.

## B.5 ADDITIONAL TEMPORAL REASONING EXPERIMENTS

To better understand the impact of temporal coherence on reasoning tasks of different models, we conducted two experiments focused on temporal reasoning. These experiments were designed to

Table 8: Performance (%) of different MLLMs across different disciplines.

| Model | Art& Sports | Business | Science | Health& Medicine | Embodied Tasks | Tech& Engineering | Average |
|---|---|---|---|---|---|---|---|
| Video-Chat (Open-ended) (Li et al., 2023c) | 27.484 | 9.091 | 18.137 | 10.417 | 29.371 | 19.149 | 22.887 |
| Video-Chat (Li et al., 2023c) | 39.355 | 48.951 | 31.863 | 45.833 | 39.161 | 38.298 | 39.588 |
| Video-LLaMA (Open-ended) (Zhang et al., 2023a) | 5.419 | 27.972 | 24.020 | 31.250 | 11.816 | 15.957 | 16.096 |
| Video-LLaMA (Zhang et al., 2023a) | 27.355 | 31.469 | 31.373 | 48.958 | 16.084 | 28.723 | 28.729 |
| ChatUnivi (Open-ended) (Jin et al., 2023) | 21.161 | 61.538 | 42.157 | 61.458 | 30.070 | 37.234 | 32.646 |
| ChatUnivi (Jin et al., 2023) | 12.387 | 58.042 | 50.000 | 60.417 | 30.070 | 43.617 | 29.072 |
| Otter (Open-ended) (Li et al., 2023a) | 37.677 | 32.867 | 37.255 | 32.292 | 22.378 | 27.660 | 34.639 |
| Otter (Li et al., 2023a) | 17.677 | 16.783 | 12.255 | 5.208 | 17.483 | 15.957 | 15.876 |
| ImageBind-LLM (Open-ended) (Han et al., 2023) | 3.355 | 3.497 | 14.706 | 10.417 | 21.678 | 18.085 | 8.179 |
| ImageBind-LLM (Han et al., 2023) | 23.742 | 34.965 | 51.471 | 33.333 | 48.951 | 56.383 | 33.952 |
| PandaGPT (Open-ended) (Su et al., 2023) | 22.581 | 16.084 | 24.020 | 21.875 | 19.580 | 21.277 | 21.718 |
| PandaGPT (Su et al., 2023) | 27.613 | 44.056 | 39.706 | 25.000 | 40.559 | 21.277 | 31.615 |
| LWM (Open-ended) (Liu et al., 2024b) | 16.000 | 20.979 | 14.706 | 16.667 | 19.580 | 20.213 | 16.976 |
| LWM (Liu et al., 2024b) | 16.387 | 18.182 | 18.137 | 19.792 | 22.378 | 21.277 | 17.938 |
| X-Instruct-BLIP (Open-ended) (Panagopoulou et al., 2023) | 3.613 | 11.888 | 14.706 | 25.000 | 17.483 | 13.830 | 9.416 |
| X-Instruct-BLIP (Panagopoulou et al., 2023) | 19.355 | 13.287 | 22.549 | 29.167 | 18.881 | 14.894 | 19.519 |

Table 9: Performance (%) of MLLMs on temporal reasoning tasks under different conditions.

| Model | Original Videos | Shuffled Videos | Reduced Video Frames |
|---|---|---|---|
| GPT-4o (OpenAI, 2024) | 40.90 | 35.11 | 32.19 |
| GPT-4V (OpenAI, 2023b) | 27.17 | 22.04 | 22.33 |
| Claude-3.5-Sonnet (Anthropic, 2024) | 25.77 | 21.58 | 19.45 |
| Gemini Pro (Team et al., 2023) | 24.65 | 20.19 | 18.97 |
| Video-LLaVA (Lin et al., 2023a) | 34.45 | 18.47 | 28.50 |
| Video-Chat-7B (Li et al., 2023c) | 25.77 | 21.50 | 20.19 |
| Video-ChatGPT-7B (Maaz et al., 2024) | 23.53 | 21.62 | 20.17 |
| ImageBind-LLM-7B (Han et al., 2023) | 19.89 | 16.19 | 14.98 |
| PandaGPT-7B (Su et al., 2023) | 28.01 | 24.35 | 22.57 |
| ChatUnivi-7B (Jin et al., 2023) | 22.97 | 19.41 | 17.14 |
| Video-LLaMA-2-13B (Zhang et al., 2023a) | 6.16 | 5.02 | 4.58 |
| X-Instruct-BLIP-7B (Panagopoulou et al., 2023) | 11.20 | 9.88 | 8.95 |
| LWM-1M-JAX (Liu et al., 2024b) | 7.00 | 5.75 | 5.56 |
| Otter-7B (Li et al., 2023a) | 9.52 | 3.25 | 7.93 |
| mPLUG-Owl-7B (Ye et al., 2023) | 20.17 | 18.19 | 16.59 |

analyze model performance under varying temporal constraints, including reduced video frames and shuffled video frames.

- **Reduced Video Frames**: Videos were processed by reducing the number of frames to 1/5 of the original. This setting evaluates the models' ability to reason with limited temporal information.
- **Shuffled Video Frames**: Videos were processed by shuffling their frames. This setting tests the models' ability to reason when the temporal order of the frames is disrupted.

The results of these experiments are summarized in Table 9. From Table 9, there is a significant performance drop when videos are either reduced in frame count or shuffled. These findings highlight the sensitivity of models to temporal coherence and emphasize the necessity of maintaining sufficient temporal information for accurate reasoning. Notably, proprietary models such as GPT-4o and GPT-4V demonstrate better resilience under these settings compared to most open-source models.

## C  IMPLEMENTATION DETAILS

We use the optimum number of video frames and report the performance in the main paper. The numbers of the sampled frames are 10 for GPT-4V/o and Gemini Pro, 8 for Video-LLaVA, 32 for ChatUniVi. For closed-source models, for both Gemini Pro and GPT-4V, we use the default settings provided by their official APIs. We use Katna [3] to extract key video frames as input to these two models. The Gemini Pro is set to process visual input and configured with safety settings to filter a range of harmful content. The configuration thresholds are set to 'BLOCK_NONE'. For

---

[3]https://github.com/keplerlab/katna

Table 10: Performance of different models across evaluations using Video-LLaVA as the evaluator.

| Model | Accuracy (%) |
|---|---|
| Video-Chat-7B (Li et al., 2023c) | 41.96 |
| ChatUnivi-7B (Jin et al., 2023) | 39.81 |
| mPLUG-Owl-7B  (Ye et al., 2023) | 38.01 |
| PandaGPT-7B (Su et al., 2023) | 31.66 |
| ImageBind-LLM-7B (Han et al., 2023) | 31.65 |
| X-Instruct-BLIP-7B (Panagopoulou et al., 2023) | 22.02 |
| LWM-1M-JAX (Liu et al., 2024b) | 16.81 |
| Otter-7B (Li et al., 2023a) | 12.08 |
| Video-LLaMA-2-13B (Zhang et al., 2023a) | 10.84 |

Table 11: Category-wise and overall error rates

| Category | Incorrect/Total | Error Rate (%) |
|---|---|---|
| Sports & Arts | 5/62 | 8.06 |
| Health & Medicine | 2/7 | 28.57 |
| Science | 1/52 | 1.92 |
| Robotics | 0/12 | 0.00 |
| Business | 0/10 | 0.00 |
| Tech & Engineering | 1/46 | 2.17 |
| **Overall** | **9/189** | **4.76** |

PandaGPT, we set 'top_p' to 0.7, and 'temperature' to 0.5. For VideoChat, we set 'max_frames' to 100. For LWM, we use the LWM-Chat-1M variant. For X-Instruct-BLIP, the model is implemented using four image frames. For Otter, we use the video variant. We use GPT-4-32K as the judge for judging whether the model answer is correct when it can not mapped to the option letter using the rule-based method. The prompt provided to GPT-4-32K is structured as follows: `"I will present a response from a question-answering model alongside several answer options. Your task is to evaluate the response and determine which of the following options it most closely aligns with, denoting the most similar option by its corresponding letter (a, b, c, or d)."`.

**Query Generation in Synthetic Data Generation Pipeline**    For the discipline of **Science**, queries are generated for subdisciplines such as Geography, Chemistry, Wildlife Restoration, Mycology, Nature, Physics, Weather, Zoology, Math, Botany, Biology, and Geology. In the **Tech & Engineering** discipline, our queries span across Electronics, Animal Behavior, Mechanical Engineering, Energy & Power, Architecture, Agriculture, Nature, Physics, Robotics, Woodworking, and Gardening. The **Sports & Arts** discipline encompasses a broad range of cultural and physical activities, including Music, Drawing and Painting, Football, Volleyball, Aerobic Gymnastics, Basketball, Instrument, Baking, Dance, Woodworking, Graffiti, Anatomy, and additional Music-related topics. **Embodied Tasks** are represented through queries for Assembly, Ego-motion, and Single Object Manipulation, focusing on the interaction between agents and their physical environment. The **Health & Medicine** discipline is segmented into Pharmacy, Public Health, Clinical Medicine, and Basic Medical Science, reflecting the multifaceted nature of healthcare and medical studies. The **Business** discipline is stratified into fundamental areas such as accounting, finance, management, marketing, and economics, each representing key facets of the commercial and economic world. Lastly, the **Game** discipline consists of Role Playing Game, First Person Shooting game, Racing Game, Adventure Game, Real-Time Strategy Game, Tower Defense game, and Fighting Game.

Each generated query retrieves relevant video content, which is then filtered and processed to align with the specific needs of our research objectives. Videos that meet our criteria in terms of content, length, and quality are downloaded and incorporated into our dataset, forming the basis for subsequent analysis and model training.

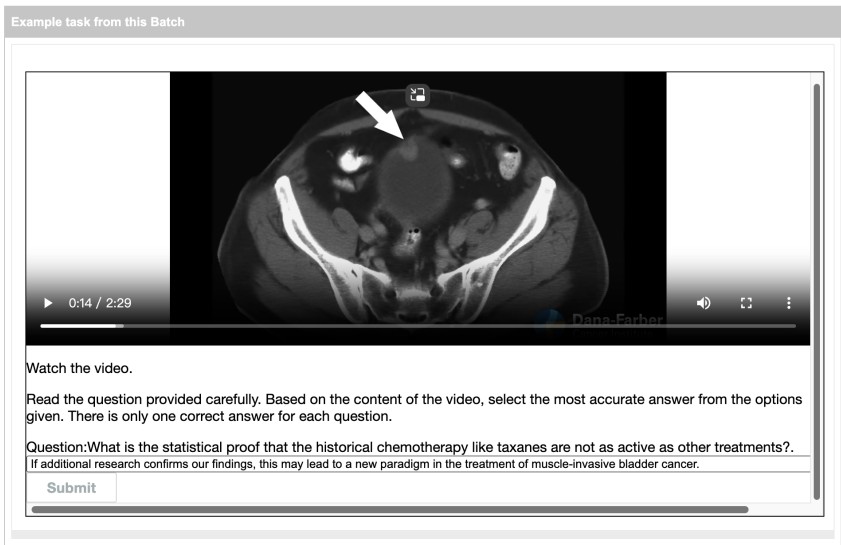

Figure 6: The interface of using Amazon Mechanical Turk to do human evaluation.

Table 12: Comparison of Human Evaluation on subset of 75 videos.

| Model | Art& Sports | Business | Science | Health& Medicine | Embodied Tasks | Tech& Engineering | Average |
|-------|-------------|----------|---------|------------------|----------------|-------------------|---------|
| Human Evaluation | 31.183 | 59.782 | 42.103 | 48.858 | 56.429 | 50.134 | 43.758 |
| GPT-4V (OpenAI, 2023b) | 30.399 | 89.203 | 68.731 | 80.059 | 38.432 | 69.108 | 48.793 |
| Gemini-Pro (Team et al., 2023) | 28.745 | 80.909 | 69.425 | 80.023 | 50.987 | 80.479 | 48.083 |

# D  HUMAN EVALUATION

## D.1  QUALITY OF DATA

We hired Amazon Mechanical Turk to do human evaluation on the data with the results shown in Table 7. Workers were required to have completed more than 1000 Human Intelligence Tasks (HITs) and have an HIT approval rate greater than 95% to qualify for our tasks. We show in Figure 6 the human evaluation interface on the generated data. Each worker was compensated 0.20 for completing an assignment. This amount was determined based on the estimated time and effort required to complete each task. We set the number of unique workers per task to 3 to collect diverse perspectives while avoiding redundancy. Workers were given 1 hour to complete each assignment. This time frame was chosen to enable thoughtful responses from workers.

We also hired students from campus to do human evaluation on subset of the data. The results are shown in Table 12. The performance of the human evaluators did not surpass that of GPT-4V and Gemini-Pro. This outcome underscores the challenging nature of the dataset, which often necessitates specialized domain knowledge that our evaluators—primarily non-experts—found demanding. These results highlight the complexity of the questions and the potential necessity for discipline-specific understanding to achieve high accuracy

## D.2  QUALITY OF USING GPT AS THE JUDGER

For a comprehensive assessment of GPT-4V's accuracy when using it as the judger, we devised a human evaluation protocol also resort to Amazon Mechanical Turk, as visualized in Figure 7. The evaluators present a series of statements derived from the video, and GPT-4V is tasked with selecting the most accurate answer from a set of multiple-choice questions. Through this interface, human evaluators can efficiently gauge GPT-4V's performance across different types of questions—when using it as the judger.

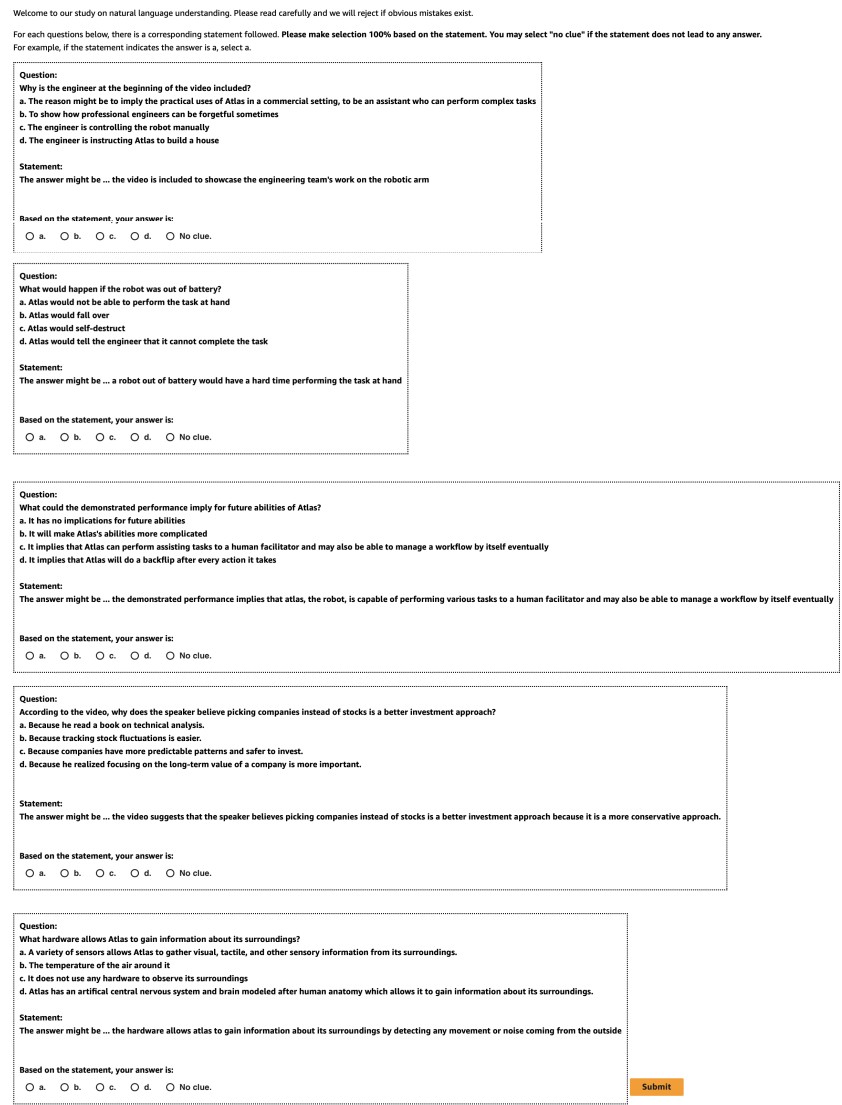

Figure 7: Human evaluation interface for GPT judger.

The results obtained from this human evaluation process are shown in Table 11, across 189 examples, there are only 9 incorrect ones with the error rate of 4.76%, validating the effectiveness of using GPT-4V as the judger.

## E  ERROR ANALYSIS

In this section, we delve into the analysis of errors from evaluated MLLMs. We summarized error types as follows:

*Question Understanding Error (QUE):*  Models misinterpret the question's intent, such as misunderstanding how a pendulum's period would change if a condition in the scenario is altered.

*Audio Understanding Error (AUE):*  Models fail to interpret audio cues correctly, shown by their failure to recognize blue and red lines on a stock chart.

*Visual Perception Error (VPE):*  There is a misinterpretation of visual content, leading to incorrect assumptions about the visual data presented in the video.

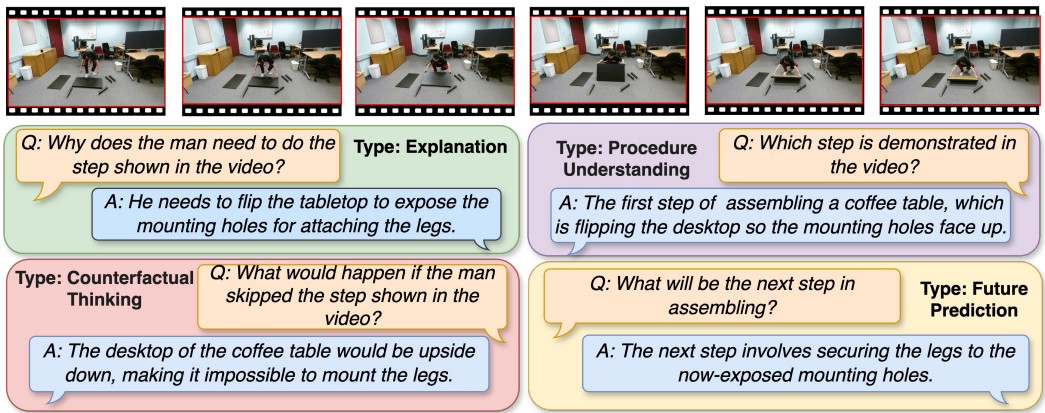

Figure 8: Examples from MMWorld in the Embodied Tasks discipline.

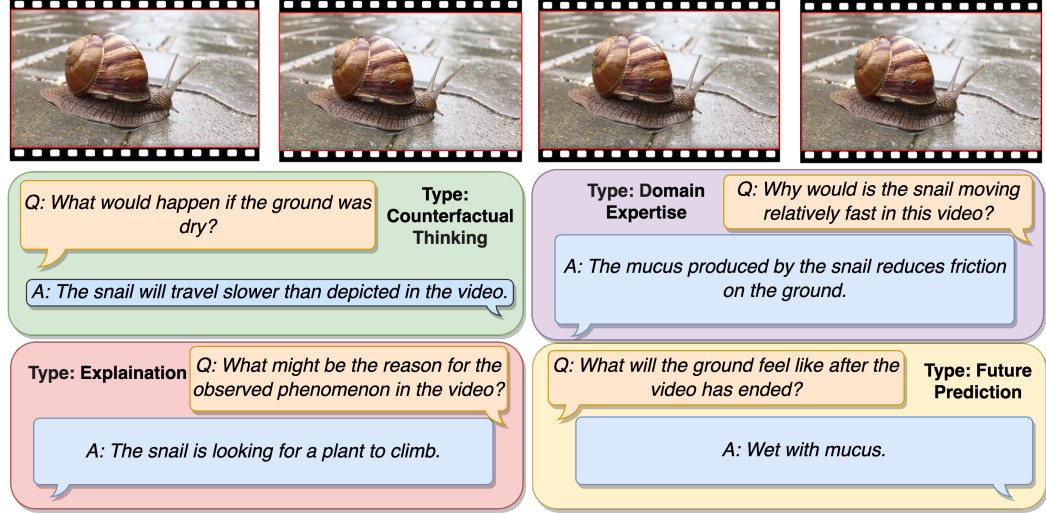

Figure 9: Examples from MMWorld in the Tech & Engineering discipline.

*Hallucinations (HE):* Models generate content or details that are not present in the actual data, essentially 'hallucinating' information.

*Reasoning Error (RE):* Models demonstrate a lack of logical reasoning, leading to incorrect conclusions based on the given data.

*Lack of Domain Knowledge (LDK):* Models show an inability to answer questions that require specific domain expertise, indicating a gap in their knowledge.

*Reject to Answer (RA):* An example of this error was observed when the model was asked to select an answer regarding the outcome of an experiment involving liquid nitrogen. Instead of choosing an option, the model provided an unrelated response concerning a light bulb, indicating either a misunderstanding or a cautious approach due to the potential for the question to be interpreted as pertaining to a sensitive topic, which can trigger content filters focused on safety and compliance policies.

We show in Figure 18, 19, 20, 21 some error cases of *Question Understanding Error*, *Audio Understanding Error*, *Visual Perception Error*, *Hallucinations*, *Reasoning Error*, *Lack of Domain Knowledge*, and *Reject to Answer* respectively from MLLMs evaluated on MMWorld.

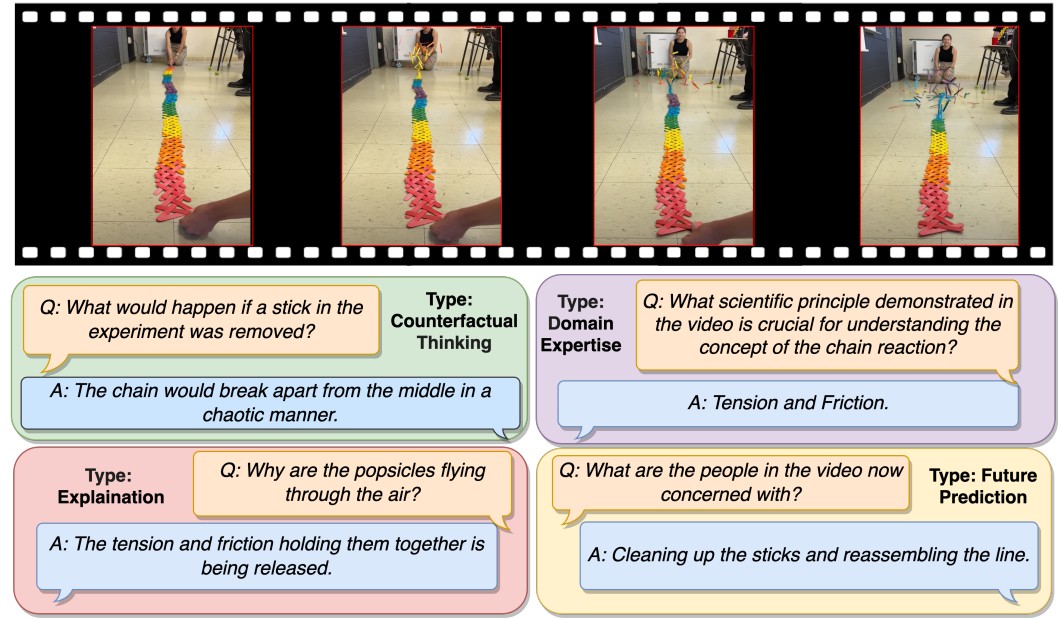

Figure 10: Examples from MMWorld in the Science discipline.

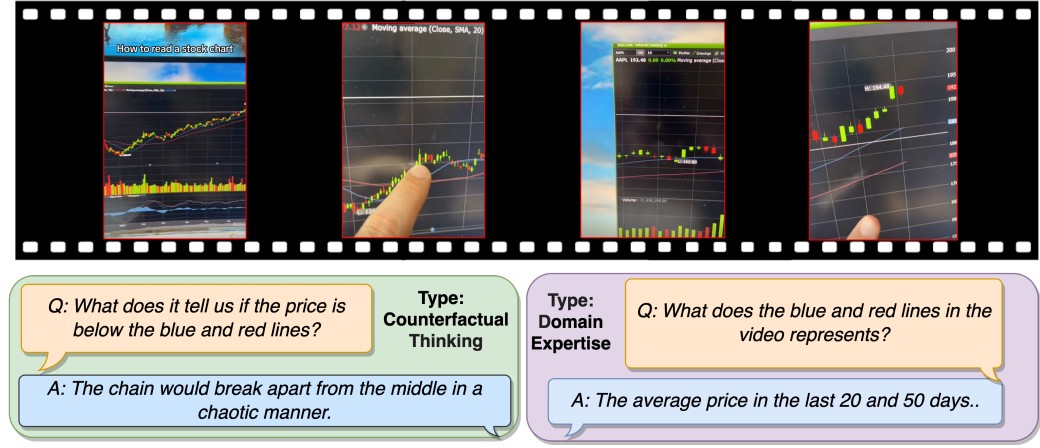

Figure 11: Examples from MMWorld in the Business discipline.

# F  DATA EXAMPLES

## F.1  MAIN SUBSET

We present additional examples from the main subset of MMWorld in Figures 8, 9, 10, 11, 12, and 13.

Furthermore, Figures 14, 15, and 16 demonstrate how Q&A pairs in MMWorld are carefully tailored to specific disciplines, including Sports & Arts, Science, and Business. Each example is designed to reflect the unique reasoning and understanding required within its respective discipline.

## F.2  SYNTHETIC I AND SYNTHETIC II

We present in Figure 17 additional examples from Synthetic I and Synthetic II of MMWorld. The examples correspond to various disciplines: Business, Health & Medicine, Science, and Gaming,

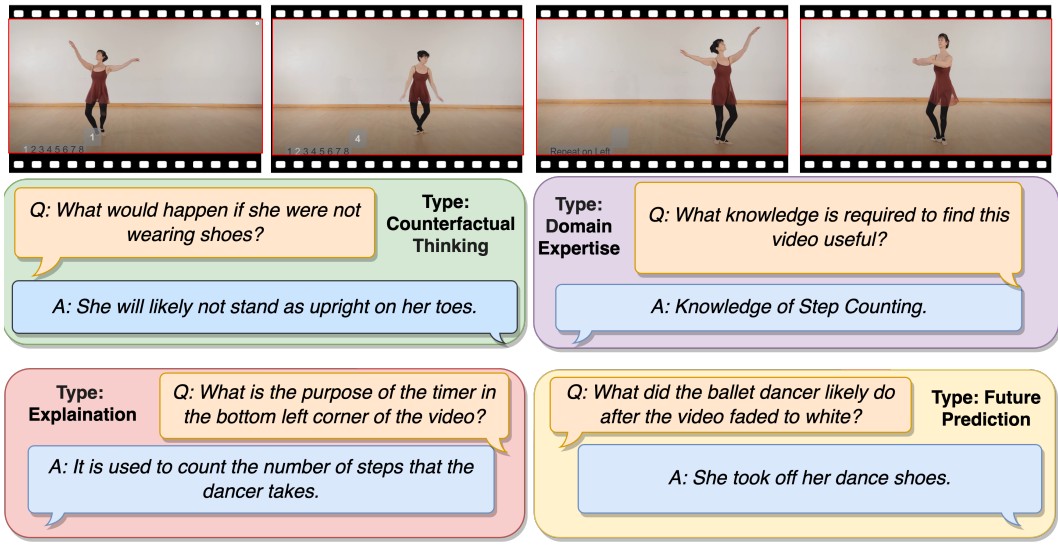

Figure 12: Examples from MMWorld in the Arts & Sports discipline.

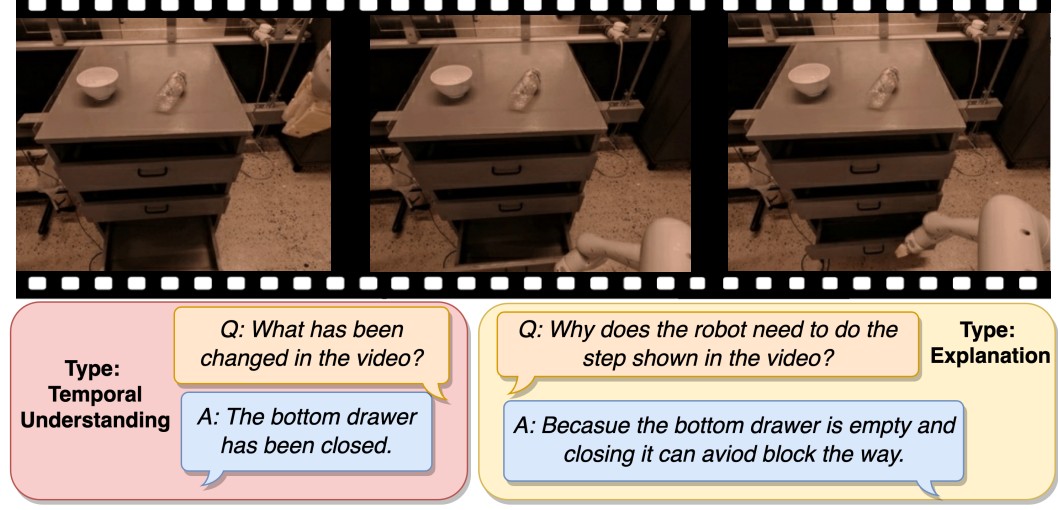

Figure 13: Examples from MMWorld of explicit temporal understanding and implicit temporal understanding (e.g., in explanation).

respectively. For each discipline, the first example showcases an audio-only generated QA from Synthetic I, while the second example represents a visual-only generated QA from Synthetic II. These examples highlight the multi-disciplinary reasoning capabilities evaluated in our benchmark, even for the synthetic dataset.

# G ADDITIONAL DATA STATISTICS

For human annotated dataset, the length of each video was capped at approximately two minutes. The statistical distribution of the disciplines within the dataset for this part is as follows:

- *Sports & Arts*: The subset that consists of 77 videos, showcasing a vibrant collection that covers a wide range of topics from athletic endeavors to various forms of artistic expression.

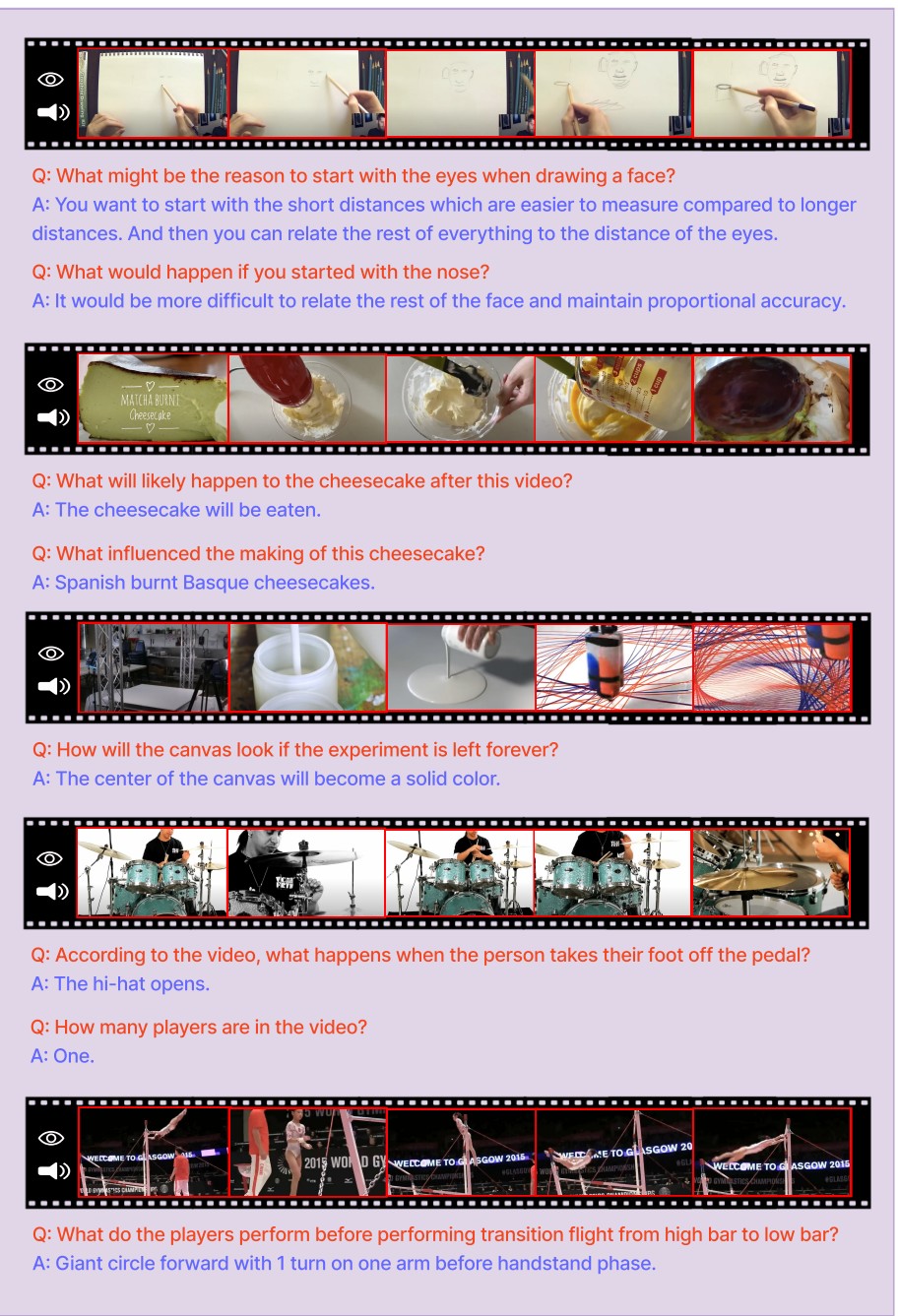

Figure 14: Examples from the Sports & Arts discipline, illustrating tailored Q&A pairs.

- *Science*: A subset of 75 videos, which delves into the empirical world of scientific inquiry, spanning a multitude of specializations from fundamental physics to advanced biological studies.

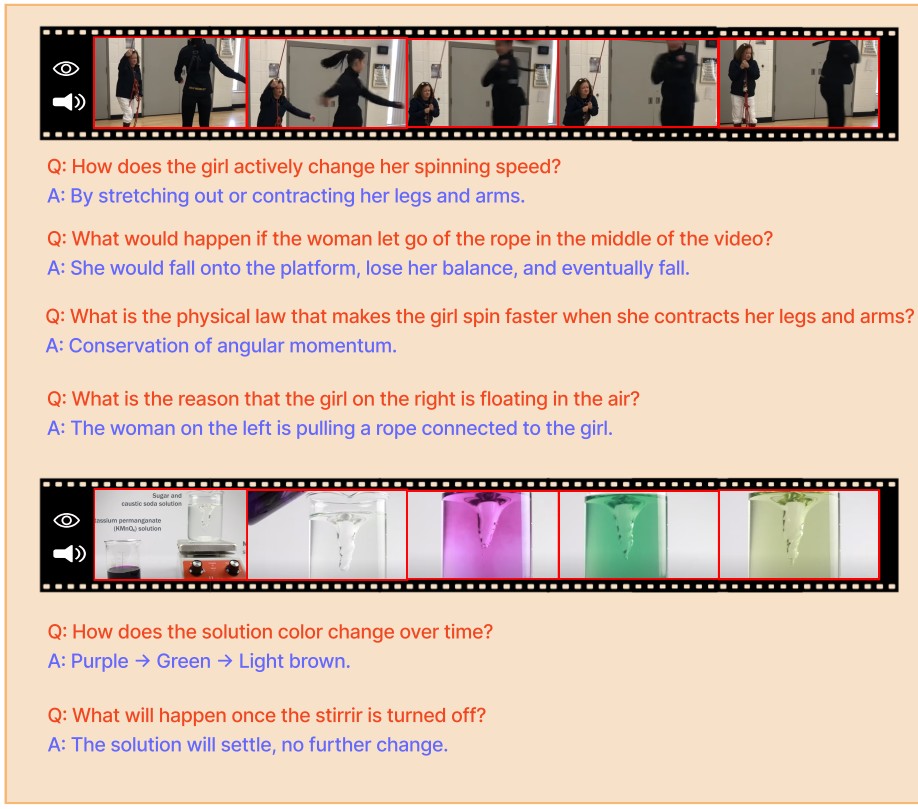

Figure 15: Examples from the Science discipline, illustrating tailored Q&A pairs.

- *Tech & Engineering*: Encompassing 54 videos, this segment captures the cutting-edge advancements and foundational concepts that drive innovation and infrastructure in the modern world.

- *Embodied Tasks*: With 50 videos, the dataset provides a focused insight into the dynamic field of Embodied Tasks, highlighting the intersection of AI, mechanics, and automation.

- *Health & Medicine*: This essential discipline is well-represented with 50 videos, offering perspectives on medical breakthroughs, healthcare practices, and life sciences.

- *Business*: This discipline includes 50 videos, reflecting on the multifaceted nature of commerce, from economics to management sciences.

- *Game*: This discipline includes 51 videos, reflecting various aspects of gaming.

Altogether, the MMWorld Benchmark's diversity is visually encapsulated in Figure 22, which delineates the distribution of videos across 61 subdisciplines. The horizontal bar chart provides a quantified representation of the dataset's range, reflecting the careful curation process that has gone into ensuring breadth across various knowledge areas.

MMWorld also has additional annotations such as "Requires Audio", "Requires Video", and "Question Only". The world we live in is rich with both audio and visual information, and effective world modeling requires an understanding of how these modalities interact and convey meaning. To achieve this, we annotated additional attributes such as "Requires Audio", "Requires Video", and "Question Only" during data collection. These annotations help determine whether correctly answering a question necessitates audio information, visual cues from the video, or can be ad-

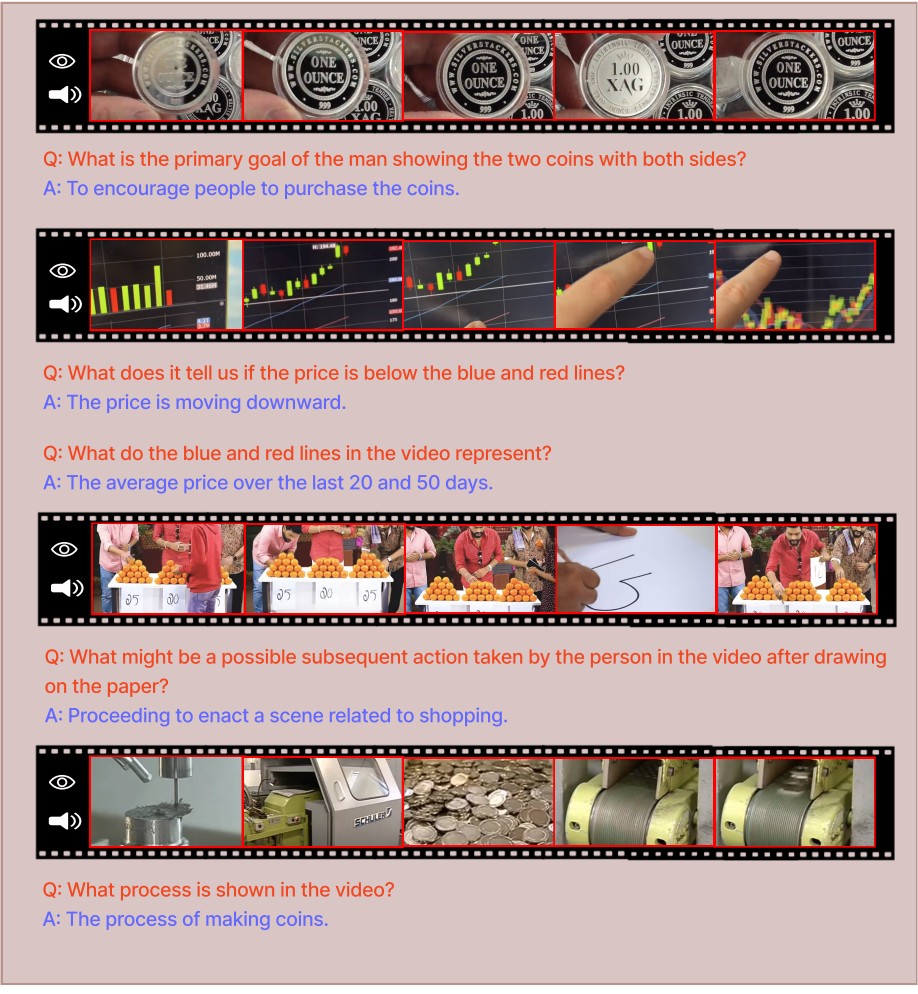

Figure 16: Examples from the Business discipline, illustrating tailored Q&A pairs.

dressed based solely on the question itself. By doing so, we ensure that our benchmark tests the full spectrum of multimodal comprehension, reflecting the complex, sensory-rich environment in which real-world understanding takes place. The statistics of these annotations are shown in Figure 23.

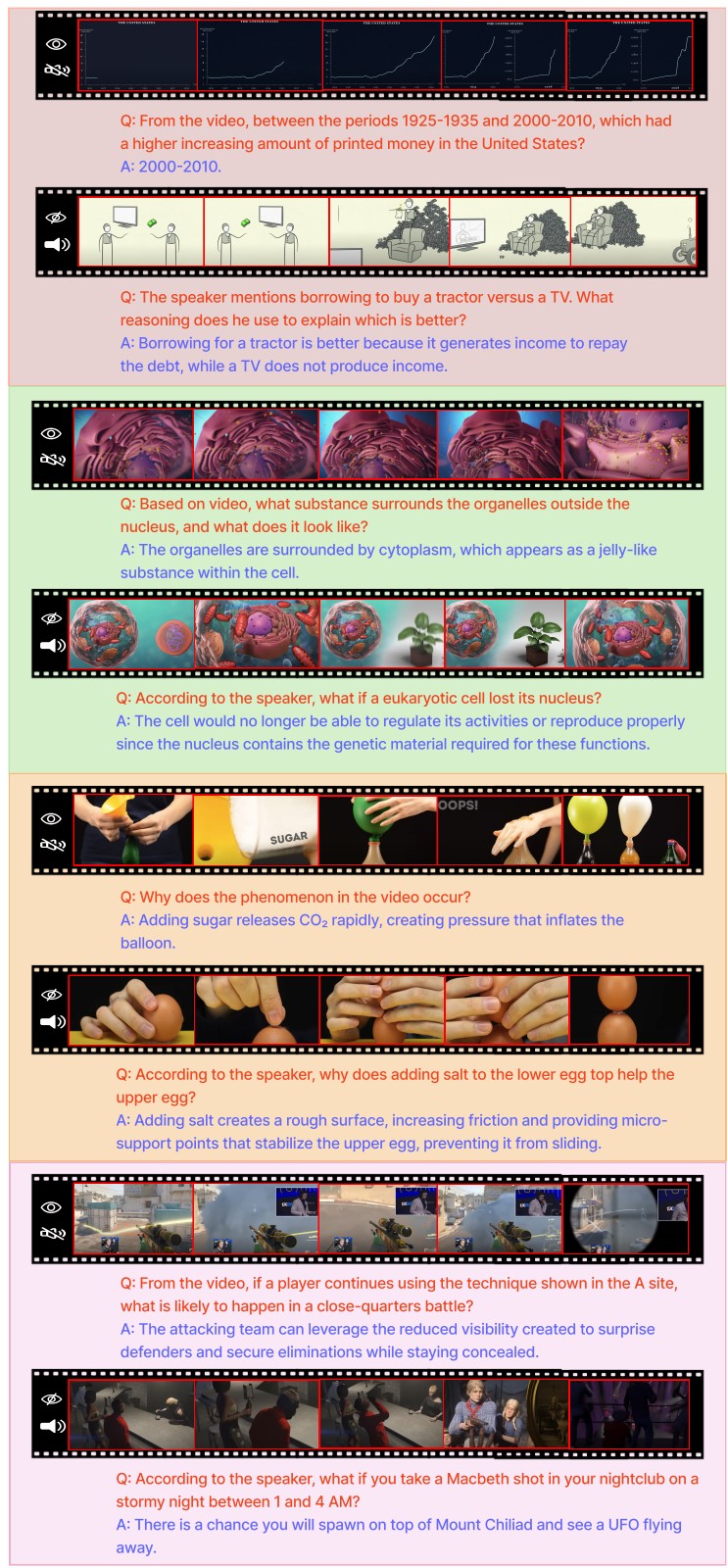

Figure 17: Examples from Synthetic Subset I and Synthetic Subset II of MMWorld.

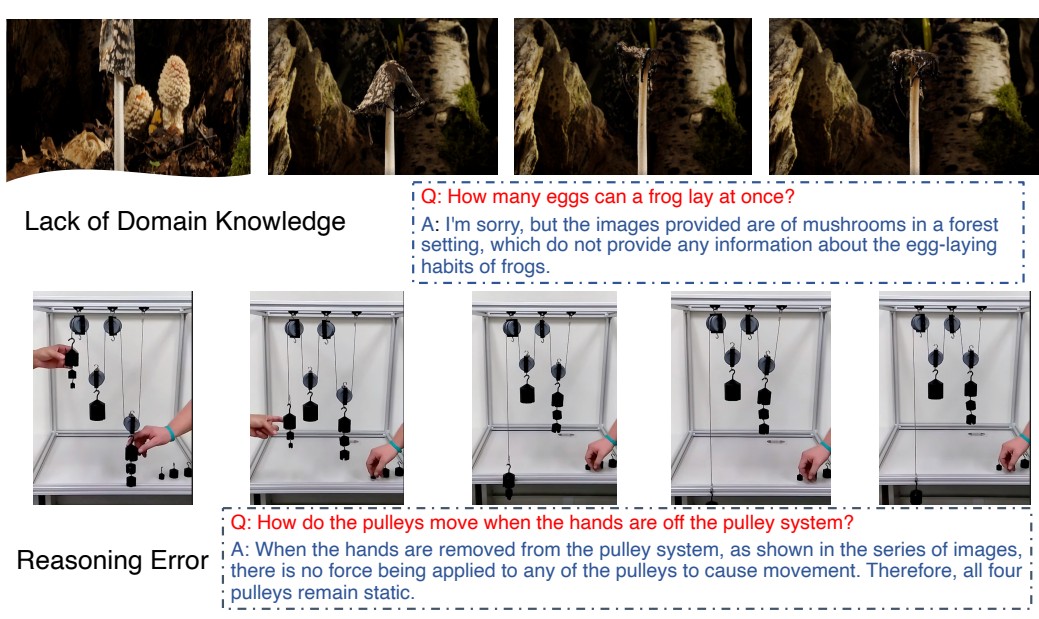

Figure 18: Error pattern of lack of domain knowledge and reasoning error. In the first case, the model does not give the correct answer because of lacking the domain knowledge. In the second case, the model makes the wrong reasoning.

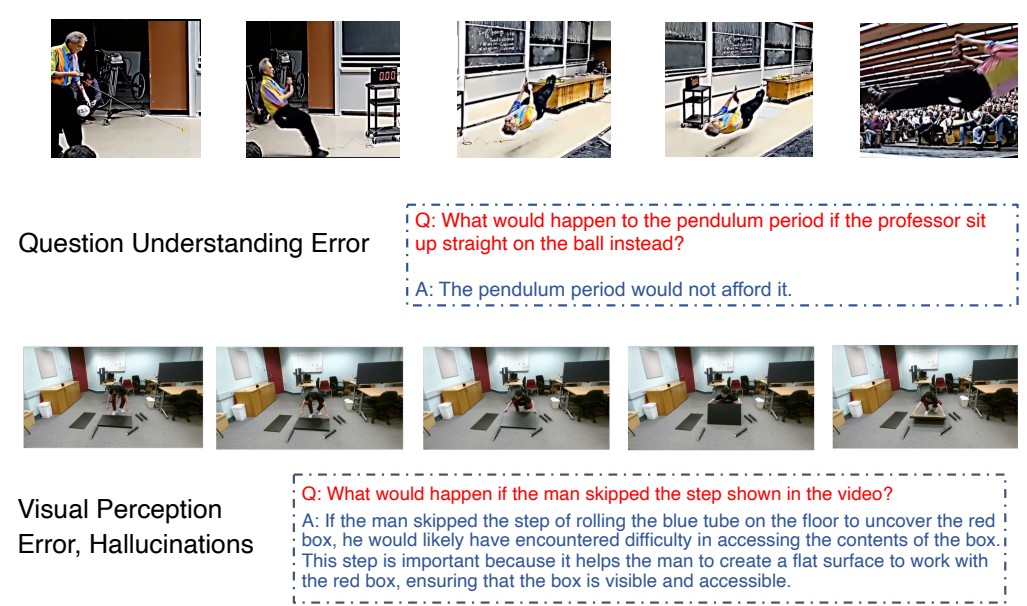

Figure 19: Error pattern of question understanding, visual perceptron error, and hallucinations. In the first case, the model does not understand the question correctly where the model fails to accurately discern the query regarding the pendulum's period. In the second scenario, the model erroneously identifies objects within the visual input, leading to the hallucination of non-existent elements, such as a red box.

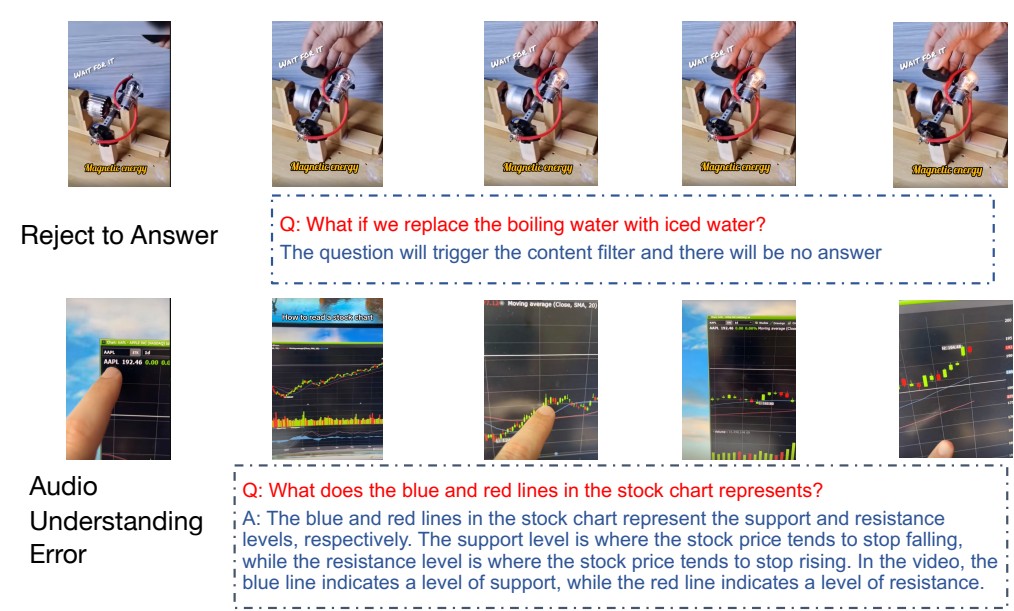

Figure 20: Error pattern of reject to answer and audio understanding error. In the first case, the model will reject to answer because of the dangerous action showing in the video. In the second case, the model does not understand the audio correctly as the answer to the question is given in the audio of the video.

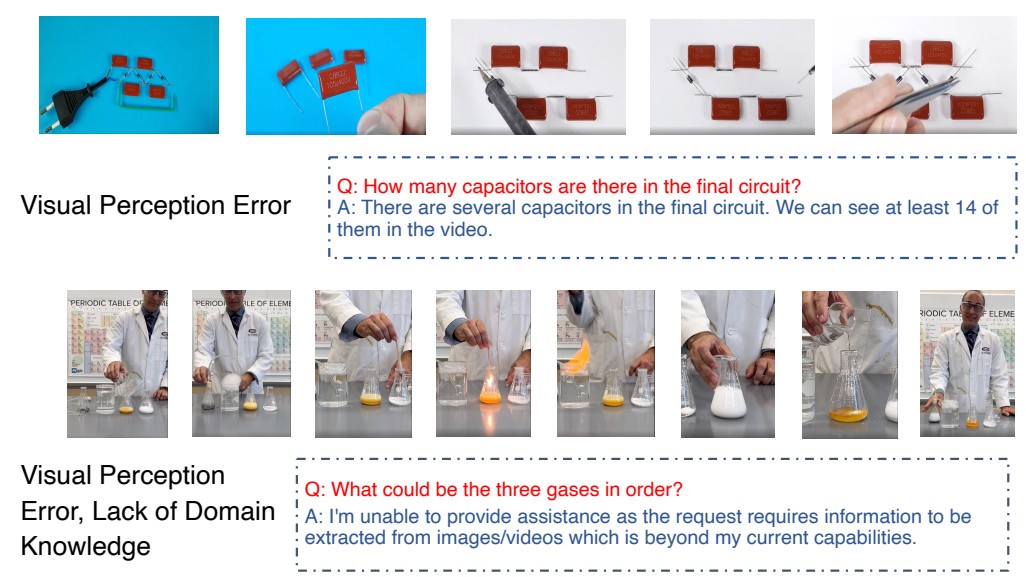

Figure 21: Error pattern due to visual perception inaccuracies and insufficient domain knowledge. The first case demonstrates a visual perception error where the model incorrectly identifies the number of capacitors present. The second case showcases a compound error where the model not only fails to discern the colors indicative of different gases but also lacks the domain knowledge necessary to infer their identity correctly.

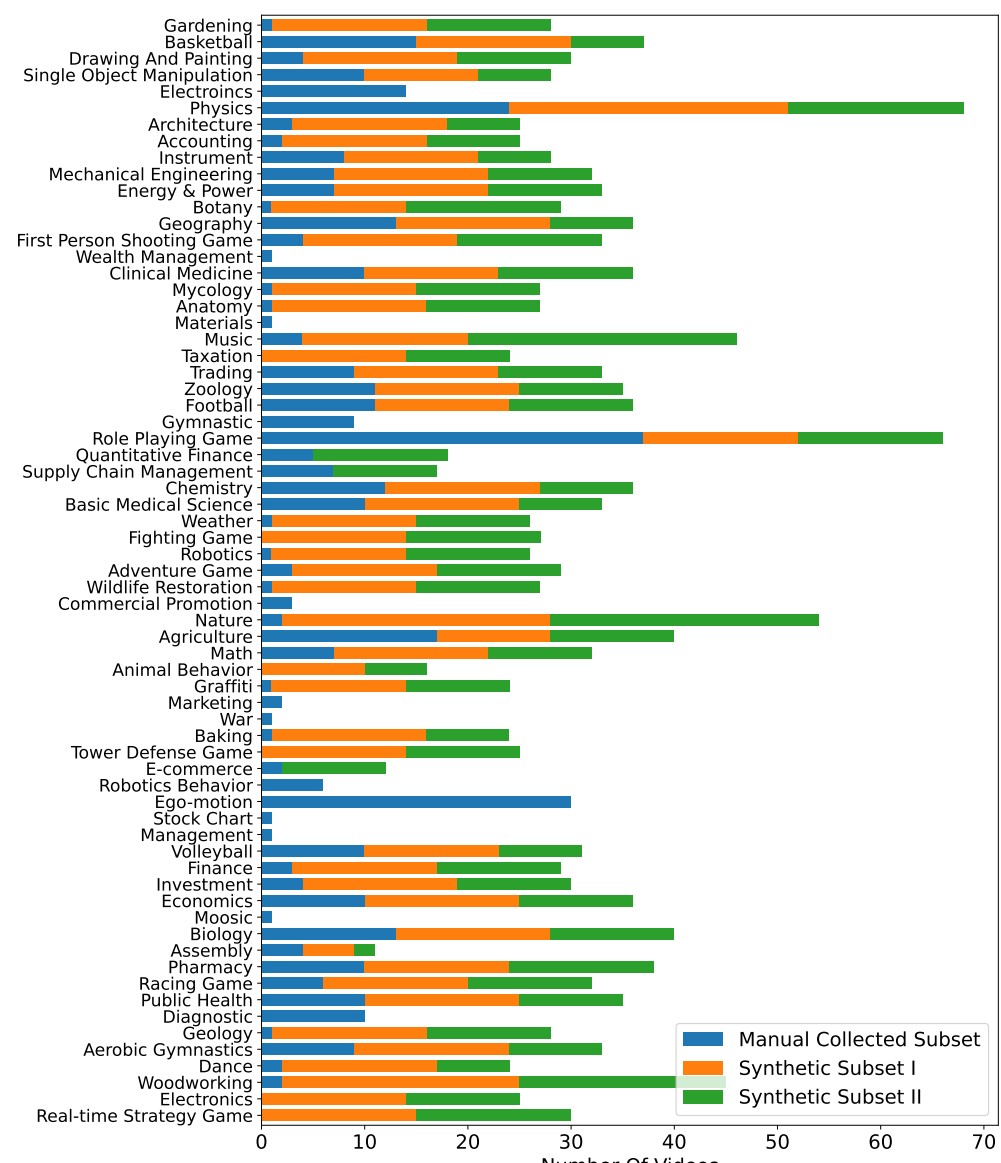

Figure 22: The number of videos per subdiscipline in MMWorld. Each horizontal bar indicates the quantity of videos corresponding to a subdiscipline, showcasing the dataset's diversity and coverage across various domains of knowledge. Synthetic Subset I is collected with audio-only data and Synthetic Subset II is collected with visual-only data.

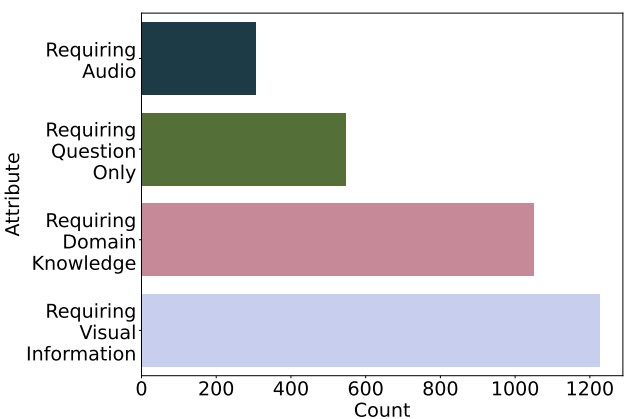

Figure 23: The distribution statistics of questions in the MMWorld benchmark by annotations.

