# OpenReview forum: "MMWorld: Towards Multi-discipline Multi-faceted World Model Evaluation in Videos"
_ICLR.cc/2025/Conference — ICLR 2025 Poster_

### Official Review · Reviewer_dopE · 2024-10-29

**Soundness:** 3
**Presentation:** 2
**Contribution:** 3
**Rating:** 6
**Confidence:** 4

**Summary:**

The paper introduces MMWorld, a new benchmark designed to evaluate multimodal large language models (MLLMs) across multiple disciplines in their understanding of real-world dynamics. MMWorld includes three datasets: a manually annotated dataset for multimodal reasoning and two automatically generated datasets focused on audio and visual understanding, respectively. Comprehensive baseline results and analyses demonstrate the challenges presented by MMWorld, with a detailed examination of primary errors and valuable insights.

**Strengths:**

1.To my best knowledge, MMWorld is the first video understanding dataset that contains diverse question types necessitating knowledge from multiple disciplines.

2.The baselines and analyses are comprehensive, covering 15 advanced MLLMs.

**Weaknesses:**

1.	According to the results in Table 3, GPT-4o already achieves high performance (80%~90%) on some disciplines, significantly surpassing other competitors. However, the QAs are generated and the model predictions are evaluated by GPT-4 families. This raises a concern that the dataset is likely biased to the knowledge of specific models (GPT-4). It would be better to show how the human annotators are instructed to amend the generated QAs.

2.	The paper seems forget to discuss and compare with two related benchmarks: NExT-QA and Causal-VidQA. The two benchmarks also emphasize explanation, temporal dynamics, counterfactual and future predictions, respectively.

**Questions:**

For the first weakness. I hope the authors can provide the followings:

1. The instructions given to human annotators for reviewing and modifying generated QAs.
2. The criteria used for accepting or rejecting generated QAs.
3. Any measures taken to ensure diversity of knowledge sources beyond GPT-4.
4. The percentage of questions that were significantly modified or entirely rewritten by human annotators.

For the second weakness. I suggest that the authors include a comparison that explicitly compares MMWorld to NExT-QA and Causal-VidQA, especially on the types of reasoning tasks.

Finally, I find that the average scores (the right most column) in Table 3 are much below than the average of the 7 disciplines. Any explanations on how to obtain the average scores?

---

> ### Author Response · Authors · 2024-11-22
> **Official Response to Reviewer dopE (1/2)**
>
> Thank you for your detailed and constructive feedback. Your comments and suggestions have provided valuable insights that helped us refine and improve our work. Below, we address each of your points in detail.
>
> ---
>
> `Weakness 1: According to the results in Table 3, GPT-4o already achieves high performance (80%~90%) on some disciplines, significantly surpassing other competitors. However, the QAs are generated and the model predictions are evaluated by GPT-4 families. This raises a concern that the dataset is likely biased to the knowledge of specific models (GPT-4). It would be better to show how the human annotators are instructed to amend the generated QAs.`
>
> We would like to clarify that all QAs in Table 3 are manually annotated by humans, not generated by GPT-4 families. We have emphasized this in the table caption for Table 3 in the revised paper.
>
> ---
>
> `Weakness 2: The paper seems forget to discuss and compare with two related benchmarks: NExT-QA and Causal-VidQA. The two benchmarks also emphasize explanation, temporal dynamics, counterfactual and future predictions, respectively.`
>
>
> `Question 2: For the second weakness. I suggest that the authors include a comparison that explicitly compares MMWorld to NExT-QA and Causal-VidQA, especially on the types of reasoning tasks.`
>
> We have added an explicit comparison with NExT-QA and Causal-VidQA to highlight the distinctions in reasoning tasks in our revision.  We also include it here:
>
> The primary distinction of MMWorld lies in its multi-discipline and multi-faceted reasoning framework, which encompasses a broader range of tasks and domains.The primary distinction of MMWorld lies in its **multi-discipline** and **multi-faceted reasoning** framework, which encompasses a broader range of tasks and domains.
> - **NExT-QA**: Focuses on advancing video understanding with multi-choice and open-ended QA tasks targeting causal action reasoning, temporal action reasoning, and scene comprehension. MMWorld, in contrast, covers a broader range of reasoning types, including:
>   1. Explanation: Elucidating the underlying logic or purpose within the video.
>   2. Counterfactual Thinking: Hypothesizing alternative outcomes.
>   3. Future Prediction: Predicting future events based on the current scenario.
>   4. Domain Expertise: Evaluating knowledge in specific fields (e.g., assembling a coffee table).
>   5. Temporal Understanding: Reasoning about temporal sequences and dynamics.
>   6. Attribution Understanding: Identifying cause-and-effect relationships, such as counting objects.
>   7. Procedure Understanding: Explaining procedural tasks demonstrated in videos.
>
> - **Causal-VidQA**: Includes four question types, focusing on scene description, evidence reasoning, commonsense reasoning, and counterfactual predictions. While Causal-VidQA emphasizes causal reasoning, MMWorld extends this scope to include **multi-discipline reasoning** across a wider variety of facets.
>
>
> We have now included a detailed comparison with NExT-QA and Causal-VidQA in the related work section of the revision.

---

> ### Author Response · Authors · 2024-11-22
> **Official Response to Reviewer dopE (2/2)**
>
> `Question 1: For the first weakness. I hope the authors can provide the followings: The instructions given to human annotators for reviewing and modifying generated QAs. The criteria used for accepting or rejecting generated QAs. Any measures taken to ensure diversity of knowledge sources beyond GPT-4. The percentage of questions that were significantly modified or entirely rewritten by human annotators.`
>
> We want to highlight that the QAs in the main subset of MMWorld are manually annotated by humans rather than being automatically generated. As mentioned in the main paper, we provided human annotators with three principles for collecting and selecting videos, which were also applied to the generation and validation of QAs:
>
> - **Multi-discipline coverage**: Videos were selected to require domain-specific knowledge across diverse disciplines.
> - **Multi-faceted annotation**: Videos were chosen to allow the creation of question-answer pairs targeting multiple reasoning facets, such as explanation, counterfactual reasoning, and future prediction.
> - **Temporal information**: Videos needed to exhibit clear temporal sequences for evaluating temporal reasoning capabilities.
>
> All QAs were reviewed by at least two professional researchers to ensure alignment with these principles. Only QAs that adhered to these standards were accepted. The dataset was further validated through human evaluations involving university students and Amazon Mechanical Turk workers to confirm quality and ensure expertise-level accuracy.
>
> These details have been clarified in the revised paper.
>
>
> ---
>
> `Question 3: Finally, I find that the average scores (the rightmost column) in Table 3 are much below the average of the 7 disciplines. Any explanations on how to obtain the average scores?`
>
> We would like to clarify that the average score in Table 3 is not calculated as a simple numerical average of the seven disciplines. Instead, it is computed as the total score across all questions divided by the total number of questions, ensuring an accurate overall average.
>
> ---
>
> We hope our responses address your concerns and clarify our contributions. Thank you for your valuable feedback, which has helped us improve the paper. Please do not hesitate to reach out if you have further questions or suggestions.

---

> ### Author Response · Authors · 2024-11-25
> **Looking forward to your response**
>
> Dear Reviewer dopE,
>
> Thank you for your valuable feedback. We have carefully addressed your comments and clarified potential misunderstandings. Additionally, we included comparisons with more related works.
>
> We kindly invite you to revisit our paper in light of these updates and hope they address your concerns and support a reevaluation of your rating.
>
> Best,
>
> The Authors

---

> > ### Comment · Reviewer_dopE · 2024-11-28
> >
> > Thank you for the clarification. I have read the revision and other reviewers' comments, and lean slightly towards acceptance.

---

> > > ### Author Response · Authors · 2024-11-28
> > >
> > > Thank you for taking the time to review our clarifications, revisions, and other reviewers’ comments. If you have any additional feedback or suggestions, please feel free to share, and we would be happy to address them. We are encouraged to hear that you lean slightly towards acceptance and would greatly appreciate it if you might consider raising your score!

---

### Official Review · Reviewer_Q3it · 2024-11-05

**Soundness:** 3
**Presentation:** 3
**Contribution:** 2
**Rating:** 6
**Confidence:** 4

**Summary:**

This paper introduces MMWorld, a new benchmark designed for multi-disciplinary, multi-faceted multimodal video understanding. MMWorld comprises 1,910 videos spanning seven broad disciplines and 69 sub-disciplines, along with 6,627 question-answer pairs and corresponding captions.

**Strengths:**

- MMWorld is the first video benchmark to cover videos beyond the general visual domain.
- The paper includes several analyses to assess the strengths and limitations of current video large language models (LLMs).

**Weaknesses:**

- Regarding multi-faceted abilities, almost all are already evaluated in previous datasets, such as CVRR [1] and Tempcompass [2]. This raises the question of the necessity for another benchmark to assess these capabilities, especially as the data in MMWorld also comes from YouTube, which may reduce its novelty and significance.

- For multi-discipline evaluation, the use of GPT-4V to generate questions raises concerns about ensuring expertise-level quality, as seen in MMMU, which uses college exam questions to maintain a rigorous standard. If the generated questions don’t meet this level, it blurs the distinction between domain-specific and general-domain questions. If no substantial difference exists, a model that performs well in general-domain benchmarks (e.g., CVRR or Tempcompass) may also perform well in MMWorld, potentially limiting the unique insights this new dataset could offer.

**Questions:**

- Regarding multi-discipline evaluation, if a model underperforms in specific domains, is it due to a lack of general domain knowledge, specific domain ability, or both? Designing experiments to analyze this would be valuable. For instance, comparative testing with models pre-trained on general knowledge versus those fine-tuned with domain-specific datasets could help isolate the impact of each knowledge type on performance.

---

> ### Author Response · Authors · 2024-11-22
> **Official Response to Reviewer Q3it (1/2)**
>
> Thank you for your detailed and constructive feedback on our submission. We appreciate your thoughtful comments and suggestions, which have been invaluable in helping us improve the clarity and rigor of our work. Below, we address each of your points in detail.
>
> `Weakness 1: Regarding multi-faceted abilities, almost all are already evaluated in previous datasets, such as CVRR [1] and Tempcompass [2]. This raises the question of the necessity for another benchmark to assess these capabilities, especially as the data in MMWorld also comes from YouTube, which may reduce its novelty and significance.`
>
> We would like to clarify that while CVRR also assesses reasoning capabilities over videos, MMWorld provides **broader coverage** in terms of **multi-disciplines** and includes more challenging reasoning abilities such as counterfactual reasoning, which CVRR does not support. Furthermore, the motivation of MMWorld differs, focusing specifically on evaluating MLLMs' ability for world modeling. It is also worth noting that CVRR is a concurrent work with ours.
>
> Similarly, TempCompass focuses solely on evaluating the temporal perception ability of Video LLMs. While we also evaluate temporal reasoning, MMWorld expands this by incorporating a wide range of reasoning facets, making **temporal reasoning just one aspect of our benchmark**. The primary distinction of MMWorld lies in its multi-discipline and multi-faceted reasoning coverage, with a much broader scope than these previous works.
>
> We have now included a detailed comparison of MMWorld with these previous benchmarks in Table 1 of the revision.
>
> Regarding the use of YouTube as a data source, we chose it because of its diverse content, which covers a wide range of disciplines and provides significant flexibility in selecting videos that meet our benchmarking principles. As outlined in the main paper, our data collection follows three strict principles: (1) multi-discipline coverage, (2) multi-faceted annotation, and (3) temporal information. This rigorous selection and annotation process required annotators to watch full videos, taking over 700 hours to complete. All videos in MMWorld are carefully selected to allow research usage and ensure copyright compliance, enhancing the significance of this dataset. Furthermore, all video-QA pairs for the main evaluation in MMWorld are manually annotated, making the collection process highly time-consuming and labor-intensive, ensuring high-quality and meaningful data.
>
> ---
>
> `Weakness 2: For multi-discipline evaluation, the use of GPT-4V to generate questions raises concerns about ensuring expertise-level quality, as seen in MMMU, which uses college exam questions to maintain a rigorous standard. If the generated questions don’t meet this level, it blurs the distinction between domain-specific and general-domain questions. If no substantial difference exists, a model that performs well in general-domain benchmarks (e.g., CVRR or Tempcompass) may also perform well in MMWorld, potentially limiting the unique insights this new dataset could offer.`
>
> We want to clarify that all data used for multi-discipline evaluation in MMWorld is annotated by humans, not generated by GPT-4V. To ensure expertise-level quality, we hired professional researchers to create questions requiring domain-specific knowledge. All annotations are cross-checked and validated by at least two independent annotators to ensure accuracy and rigor.
>
> Additionally, we performed human evaluations by hiring university students and using three independent sets of Amazon Mechanical Turk workers to validate the quality of the data. These evaluations confirm the expertise-level quality of the annotated dataset.
>
> **The GPT-4V-generated data is only used in subsets I and II**, which are specifically designed for ablation studies and are not part of the main multi-discipline evaluation dataset.

---

> ### Author Response · Authors · 2024-11-22
> **Official Response to Reviewer Q3it (2/2)**
>
> `Question 1: Regarding multi-discipline evaluation, if a model underperforms in specific domains, is it due to a lack of general domain knowledge, specific domain ability, or both? Designing experiments to analyze this would be valuable. For instance, comparative testing with models pre-trained on general knowledge versus those fine-tuned with domain-specific datasets could help isolate the impact of each knowledge type on performance.`
>
> Thank you for this insightful suggestion. We have conducted additional experiments to analyze this question by fine-tuning a model pre-trained on general knowledge with domain-specific datasets.
>
> For these experiments, we selected Video-LLaVA-7B, as it performed the best among all open-source MLLMs in our evaluation. Video-LLaVA-7B is pre-trained on extensive general knowledge datasets, including 558K LAION-CCSBU image-text pairs and 702K video-text pairs from WebVid.
>
> We observed that Video-LLaVA underperforms significantly in the Health & Medicine domain, achieving only 32.64% accuracy. After fine-tuning the model for two epochs on the PathVQA and MedVidQA datasets, its performance in the Health & Medicine domain increased to 38.54%, demonstrating the effectiveness of domain-specific fine-tuning. The results (%) are summarized below:
>
> | Discipline         | Before Fine-Tuning | After Fine-Tuning |
> |--------------------|---------------------|--------------------|
> | Art & Sports       | 35.91              | 35.48             |
> | Business           | 51.28              | 51.20             |
> | Science            | 56.30              | 57.77             |
> | Health & Medicine  | 32.64              | 38.54             |
> | Embodied Tasks     | 63.17              | 63.98             |
> | Tech & Engineering | 58.16              | 59.56             |
> | Game               | 49.00              | 49.74             |
>
> These results show that fine-tuning improves performance in the specific domain (Health & Medicine) while having minimal impact on other disciplines. This suggests that a lack of domain-specific knowledge is a key factor in underperformance for specific domains, highlighting the importance of both general and domain-specific knowledge for robust performance across disciplines.
>
> ___
>
> We hope our responses and the updates to the paper address your concerns and clarify the contributions of our work. We sincerely appreciate the time and effort you have taken to review our submission and provide thoughtful feedback. Please feel free to reach out with any further questions or suggestions. Thank you!

---

> ### Author Response · Authors · 2024-11-25
> **Looking forward to your response**
>
> Dear Reviewer Q3it,
>
> Thank you for your valuable feedback. We have clarified MMWorld’s unique contributions compared with previous works, addressed potential misunderstandings, and added experiments analyzing general versus domain-specific knowledge.
>
> We kindly invite you to revisit our paper in light of these updates. We hope these improvements can address your concerns and encourage a reevaluation of your rating.
>
> Best,
>
> The Authors

---

> > ### Comment · Reviewer_Q3it · 2024-11-26
> >
> > Thank you for your response. Based on the description in the paper, the QA dimensions mentioned in MMVU (e.g., explain, future, domain, etc.) are already reflected in existing benchmarks such as VideoMME and CVRR. For instance, VideoMME covers dimensions like explain, future, and domain, while CVRR includes explain, future, and counterfactual. Therefore, if MMVU does not demonstrate significant differences from VideoMME and CVRR in the video domain, its importance may appear somewhat limited.
> >
> > Specifically in the video domain, although the authors emphasize the multi-discipline feature, these dimensions can already be found in existing VideoQA datasets. For example, fields like Art & Sports, Health & Medicine, Tech & Engineering, and Games are present in VideoMME or CVRR, though they were not explicitly "highlighted" in these datasets. Based on this understanding, the authors may need to provide objective metrics to demonstrate MMVU's uniqueness in terms of video diversity, rather than simply listing potential dimensions. For instance, showing that MMVU encompasses more diverse videos or significantly exceeds the video volume of VideoMME (900 videos) and CVRR (217 videos) could further substantiate its diversity and advantages.

---

> ### Author Response · Authors · 2024-11-26
> **Additional Clarification and Demonstration of Our Uniqueness**
>
> Dear Reviewer Q3it,
>
> Thank you again for your follow-up comments and suggestions! We will clarify our unique contributions and advantages further to address your concerns.
>
> - First, we would like to respectfully reiterate that both CVRR and VideoMME are concurrent works with us and have not yet been officially accepted. For instance, CVRR is also another ICLR submission.
>
>
> - Second, our benchmark offers significantly greater diversity in disciplines compared to both VideoMME and CVRR. It spans seven disciplines and 69 subdisciplines—**2.3 times more extensive than VideoMME** (30 sub-class video types) and **6.3 times larger than CVRR** (11 video dimensions), both of which do not explicitly define categories by disciplines. Additionally, **Our questions are manually annotated to align with these specific disciplines**. In contrast, VideoMME organizes its videos into six general visual domains (e.g., Knowledge, Film & Television, Sports Competition, Life Record, Multilingual), which are not discipline-based. Annotators in VideoMME create questions after watching videos but are not required to target specific disciplines. Similarly, CVRR uses GPT-3.5 to generate QAs without explicitly aligning them with disciplines, limiting its ability to evaluate discipline-specific reasoning. Neither benchmark could evaluate MLLMs’ reasoning capabilities across disciplines as we do.
>
> - Third, our benchmark surpasses both of these benchmarks in dataset size. It features 1,910 videos and 6,627 QAs, which is **over twice the video volume of VideoMME** (900 videos) and nearly **nine times that of CVRR** (217 videos). The QA count is also **2.5 times larger than VideoMME** (2,700 QAs) and **2.8 times larger than CVRR** (2,400 QAs).
>
> - Fourth, our benchmark provides greater diversity in question types, covering seven QA dimensions, such as explanation, counterfactual thinking, and temporal reasoning. They are manually annotated, ensuring high quality and diversity in question patterns. In contrast, CVRR's reliance on GPT-3.5 for automatic QA generation may result in less varied question patterns.
>
>
> - Finally, our benchmark provides more comprehensive and diverse evaluation toolkit options compared to both benchmarks. It incorporates **both** LLM-based evaluation (using GPT-4V and Video-LLaVA as evaluators) and rule-based accuracy evaluation via the LLMEvals toolkit. In contrast, CVRR relies solely on LLM-based evaluation, while VideoMME is limited to rule-based accuracy evaluation.
>
> We hope these clarifications address your concerns and encourage a reconsideration of your rating. Thank you again for your valuable comments and thoughtful engagement!
>
> Best,
>
> The Authors

---

> > ### Comment · Reviewer_Q3it · 2024-11-26
> >
> > Thank you for your response.
> >
> > "Our questions are manually annotated to align with the specific requirements of each discipline." Could you provide additional examples of how to annotate Q&A pairs tailored to specific disciplines? Specifically, I would like examples of seven primary understanding and reasoning abilities Q&A pairs for each of the following domains: Sports & Arts, Science, and Business. Since this is a benchmark, it is crucial to ensure that the annotations in this paper are of high quality and that the questions are designed to distinctly reflect the unique characteristics of each discipline, rather than resembling those of general domains.

---

> > > ### Author Response · Authors · 2024-11-27
> > > **Additional Examples (1/2)**
> > >
> > > Dear reviewer Q3it,
> > >
> > > Thank you for your further questions! We have provided examples of Q&A pairs tailored to the three mentioned domains within the seven primary understanding and reasoning abilities. These examples have been included in the revision, with the corresponding video figures available in Appendix F (Figures 14, 15, and 16).
> > > Below, we also shows how each QA pair is carefully tailored to reflect discipline-specific reasoning and understanding.
> > >
> > > ### **Sports & Arts**
> > > - **Explanation:**
> > > > **Q**: *What might be the reason to start with the eyes when drawing a face?*
> > >   **A**: *To measure short distances first, which are easier to gauge, and then relate the rest of the proportions to the distance of the eyes.*
> > >
> > >   This question targets the reasoning process specific to drawing, a key skill in the Arts discipline.
> > > - **Counterfactual Thinking:**
> > > > **Q**: *What would happen if you started with the nose?*
> > >   **A**: *It would be more difficult to relate the rest of the face and maintain proportional accuracy.*
> > >
> > >   This question targets how starting points influence proportionality, a concept central to artistic creation.
> > >
> > > - **Future Prediction:**
> > > > **Q**: *How will the canvas look if the experiment is left forever?*
> > >   **A**: *The center of the canvas will be a solid color.*
> > >
> > >   This question ties directly to the culinary arts in food preparation.
> > >
> > > - **Domain Expertise:**
> > > > **Q**: *What influenced the making of this cheesecake?*
> > >   **A**: *Spanish burnt Basque cheesecakes.*
> > >
> > >   This question shows specialized knowledge of culinary history and technique, an aspect of expertise within the Arts discipline.
> > >
> > > - **Temporal Understanding:**
> > > > **Q**: *What do the players perform before performing transition flight from high bar to low bar?*
> > >   **A**: *Giant circle forward with 1 turn on one arm before handstand phase.*
> > >
> > >   This question is tailored to the Sports discipline, particularly gymnastics, as it focuses on a technical sequence performed in a high bar-to-low bar transition
> > >
> > > - **Attribution Understanding:**
> > > > **Q**: *How many players are in the video?*
> > >   **A**: *One.*
> > >
> > >   This evaluates the recognition of player in the performing arts.
> > >
> > > - **Procedure Understanding:**
> > > > **Q**: *According to the video, what happens when the person takes their foot off the pedal?*
> > >   **A**: *The hi-hat opens.*
> > >
> > >   This question relates to musical instrumentation in the performing arts.
> > > ---
> > > ### **Science**
> > > - **Explanation:**
> > > > **Q**: *How does the girl actively change her spinning speed?*
> > >   **A**: *By stretching out or contracting her legs and arms.*
> > >
> > >   This question focuses on a scientific principle—conservation of angular momentum—showcasing reasoning unique to physics.
> > >
> > > - **Counterfactual Thinking:**
> > > > **Q**: *What would happen if the woman let go of the rope in the middle of the video?*
> > >   **A**: *She would fall onto the platform, lose her balance, and eventually fall.*
> > >
> > >   This question asks reasoning about alternative physical scenarios, rooted in the scientific principles of force and balance.
> > >
> > > - **Future Prediction:**
> > > > **Q**: *What will happen once the stirrir is turned off?*
> > >   **A**: *The solution will settle, no further change.*
> > >
> > >   This question evaluates the ability to predict the outcome of chemical or physical processes, central to scientific reasoning.
> > >
> > > - **Domain Expertise:**
> > > > **Q**: *What is the physical law that makes the girl spin faster when she contracts her legs and arms?*
> > >   **A**: *Conservation of angular momentum.*
> > >
> > >   This question evaluates specialized knowledge in physics discipline.
> > >
> > > - **Temporal Understanding:**
> > > > **Q**: *How does the girl actively change her spinning speed?*
> > >   **A**: *By stretching out or contracting her legs and arms.*
> > >
> > >   This question is also a temporal understanding question which relates to understanding of temporal dynamic processes over time, a key aspect of physical science.
> > >
> > > - **Attribution Understanding:**
> > > > **Q**: *What is the reason that the girl on the right is floating in the air?*
> > >   **A**: *The woman on the left is pulling a rope connected to the girl.*
> > >
> > > This question is tailored to science as it tests understanding of force and tension, focusing on the causal mechanics of a rope system.
> > >
> > > - **Procedure Understanding:**
> > > > **Q**: *How does the solution color change over time?*
> > >   **A**: *Purple → Green → Light brown.*
> > >
> > >   This evaluates procedural understanding of time-dependent chemical reactions, specific to experimental sciences.

---

> ### Author Response · Authors · 2024-11-27
> **Additional Examples (2/2)**
>
> ### **Business**
>
> - **Explanation:**
> > **Q**: *What is the primary goal of the man showing the two coins with both sides?*
>   **A**: *To encourage people to purchase the coins.*
>
>   This question relates to marketing strategies and consumer behavior, core elements of business reasoning.
>
> - **Counterfactual Thinking:**
> > **Q**: *What does it tell us if the price is below the blue and red lines?*
>   **A**: *The price is moving downward.*
>
>   This question evaluates understanding of financial data visualization, specific to business analytics.
>
> - **Future Prediction:**
> > **Q**: *Based on the video, what can be predicted about the actions of governments, banks, and companies in response to the condition of the gross domestic product?*
>   **A**: *They will act based on the trends indicated for the next accounting year.*
>
>   This questions asks for economic forecasting and strategic planning, essential skills in the business discipline.
>
> - **Domain Expertise:**
> > **Q**: *What do the blue and red lines in the video represent?*
>   **A**: *The average price over the last 20 and 50 days.*
>
>   This question asks for knowledge of financial indicators, directly tied to domain expertise in economics and business.
>
> - **Temporal Understanding:**
> > **Q**: *What might be a possible subsequent action taken by the person in the video after drawing on the paper?*
>   **A**: *Proceeding to enact a scene related to shopping.*
>
>   This question is tailored to temporal reasoning about sequential business actions.
>
> - **Attribution Understanding:**
> > **Q**: *According to the video, what is a key factor in movement towards more efficient allocation of capital globally?*
>   **A**: *Allocation of capital based on what each country produces best*
>
>   This question is tailored to asking about understanding of international trade principles in the business discipline.
>
> - **Procedure Understanding:**
> > **Q**: *What process is shown in the video?*
>   **A**: *The process of making coins.*
>
>   This question captures the step-by-step procedure of production, essential to operations management in business.
> ---
> We hope this additional clarification addresses your concerns and could encourage a reconsideration of the rating. Thank you!
>
> Best,
>
> Authors

---

> > ### Comment · Reviewer_Q3it · 2024-11-27
> >
> > Thank you for your response.
> >
> > Based on the examples provided, my main concern—that the question is not discipline-specific—has been partially addressed. Considering this, I have decided to raise my score.

---

> > > ### Author Response · Authors · 2024-11-27
> > >
> > > Thank you for taking the time to review our responses and updates. Your suggestions and comments have also been helpful in improving our work. We also greatly appreciate your increased rating!

---

### Official Review · Reviewer_ZqHq · 2024-11-05

**Soundness:** 3
**Presentation:** 3
**Contribution:** 3
**Rating:** 6
**Confidence:** 4

**Summary:**

This work proposes a novel benchmark to evaluate commonsense knowledge and reasoning capabilities of the existing multimodal LLMs that are capable of processing videos or sequences of images. The proposed benchmark is classified as *multi-discipline* since it includes videos from a diverse set of subjects, and it is also classified as *multi-faceted* because it also includes different types of tasks where some tasks are focusing on commonsense reasoning abilities. The authors also collected a synthethic dataset to evaluate either auditory or vision processing capabilities of these models. The experiments reveal that (i) most of the existing models fail to excel in the proposed benchmark, (ii) even the proprietary models lack deficiency in embodied tasks, (iii) and they fail to process auditory information sufficiently.

**Strengths:**

I think the most valuable aspect of this paper, the proposed dataset contains videos from a diverse set of subjects, and questions have a variety focusing on commonsense knowledge and reasoning. This unique combination could position the benchmark as an important resource for the near future, as recent models fails to thrive in this video question answering benchmark.

**Weaknesses:**

- Presentation could be improved (see the questions field).
- The paper actually has a main contribution, which is human annotated dataset. There are also two synthetic datasets to evaluate the performance of individual modality processing performances of the models that are capable of processing both vision and audio. This aspect complicates this work a little bit, where I think it would better to have a separate work which carefully examines how well the models process each modality.
- The related work could be improved there are a lot of recent related work which are missing (e.g. [TempCompass](https://arxiv.org/abs/2403.00476), [VILMA](https://openreview.net/forum?id=liuqDwmbQJ), [VITATECS](https://arxiv.org/abs/2311.17404v1), also older but related since nearly half of the questions focus on explainability [Next-QA](https://arxiv.org/abs/2105.08276))
- The imbalanced nature of the subtasks: 48% of the questions belong to the explanation category/task. Also, the main task figure gives the impression of that these tasks could be fluid. For instance, the *explanation* example also requires *spatiotemporal reasoning*, and *future prediction* requires also some *domain-expertise*. Maybe, rather than assigning just single category, they could have multiple categories.

**Questions:**

- The hexagon figure (Fig.4) is really difficult to interpret: I think this kind of figures makes sense if one compares 3 models at most. The score normalization makes things even worse: So we cannot see how well the best model performs in this way. I would definitely remove it from the paper, and put the Table 8 into the main text. Also note that the figure is embedded as "PNG/JPEG" instead of "PDF" (but removing it from the paper would make things much more better).
- Fig. 6 is also hard to interpret. I would rather group some selected models instead of error categories, so one can observe which model produces what kind of errors more or less.
- Is there any example for auditory questions in the appendix? Can you please point? Can we see some synthethic questions/answer pairs generated using vision and audio side-by-side?
- How much diverse is the language, i.e. questions and answers?
- One suggestion: Please embed all figures as PDFs.
- It would be good to see many more examples in the appendix in a more compact format by avoiding large text with conversation bubbles.

**Details Of Ethics Concerns:**

N/A.

---

> ### Author Response · Authors · 2024-11-22
> **Official Response to Reviewer ZqHq (1/2)**
>
> `Weakness 1: Presentation could be improved (see the questions field).`
>
> We have improved the presentation of our paper and uploaded a new revision, addressing the points raised in the questions field.
>
> ---
>
> `Weakness 2: The paper actually has a main contribution, which is human-annotated dataset. There are also two synthetic datasets to evaluate the performance of individual modality processing performances of the models that are capable of processing both vision and audio. This aspect complicates this work a little bit, where I think it would be better to have a separate work which carefully examines how well the models process each modality.`
>
> We would like to clarify and emphasize that the synthetic datasets are an essential component of MMWorld and play a critical role in world modeling by enabling the analysis of MLLMs within a single modality of perception. These datasets allow us to evaluate how well MLLMs can perceive and reason about the world when limited to either the audio or visual modality, which is a key aspect of their capability assessment. They also allow ablation studies to isolate and evaluate each modality's contributions to reasoning performance. To serve as a comprehensive benchmark for world modeling, the ability to process both vision and audio is indispensable. Rather than being treated as separate contributions, these datasets are integral to MMWorld and are designed to complement the human-annotated dataset.
>
> Moreover, the synthetic datasets are annotated with disciplines and reasoning facets consistent with the main dataset, enabling a unified evaluation framework. This ensures that model performance can be assessed across different disciplines and reasoning facets, even within individual modalities.
>
> Meanwhile, videos inherently combine multiple modalities, such as audio and visual components, and understanding the impact of each is vital for evaluating multimodal reasoning. Including these synthetic datasets within MMWorld is therefore crucial for our unified and comprehensive evaluation approach.
>
> ___
>
> `Weakness 3: The related work could be improved there are a lot of recent related work which are missing (e.g. TempCompass, VILMA, VITATECS, also older but related since nearly half of the questions focus on explainability Next-QA).`
>
> We appreciate the suggestion to expand the related work section. The revised version now includes detailed comparisons with the mentioned works:
>
> - **TempCompass** focuses on evaluating the temporal perception abilities of Video LLMs. However, it lacks multifaceted reasoning such as explanation and counterfactual reasoning, and it does not include multi-discipline reasoning. In contrast, our benchmark covers temporal reasoning as one of many facets while also incorporating broader reasoning abilities across multiple disciplines.
>
> - **VILMA** also evaluates temporal reasoning abilities of MLLMs, showing that current VLMs' grounding capabilities are no better than those of vision-language models that use static images. However, it does not involve multi-discipline or multifaceted reasoning as we do.
>
> - **VITATECS** is a diagnostic Video-Text dataset focused on evaluating Temporal Concept Understanding. While it specifically targets this area, our benchmark includes temporal reasoning as one facet alongside other reasoning tasks, providing a broader evaluation framework.
>
> - **Next-QA** is a rigorously designed video question-answering benchmark emphasizing causal and temporal action reasoning. However, it does not provide a multi-discipline, multifaceted reasoning framework like ours. Additionally, it includes a training set and is designed for advancing video understanding rather than purely for evaluation, which is the core focus of our benchmark.
>
> The primary distinction between our work and these benchmarks lies in our focus on multi-discipline, multifaceted reasoning, covering a wide array of disciplines and reasoning types. Temporal reasoning is just one facet in our comprehensive evaluation framework. Furthermore, MMWorld is specifically designed to evaluate MLLMs' ability for world modeling, a unique motivation not shared by these other works.
>
> The detailed comparisons have been made explicit in the revision.

---

> ### Author Response · Authors · 2024-11-22
> **Official Response to Reviewer ZqHq (2/2)**
>
> `Weakness 4: The imbalanced nature of the subtasks: 48% of the questions belong to the explanation category/task. Also, the main task figure gives the impression of that these tasks could be fluid. For instance, the explanation example also requires spatiotemporal reasoning, and future prediction requires also some domain expertise. Maybe, rather than assigning just single category, they could have multiple categories.`
>
> > Explanation questions form a large portion of our benchmark (48%) because they are crucial for evaluating a model's causal reasoning and thinking abilities, which are key to world modeling. Among all multifaceted question types in the benchmark, explanation questions are particularly challenging for existing models and are essential for understanding a model's reasoning capabilities.
>
> > Thanks for the suggestion!
> >  - We think it is a great idea for some questions to fall into multiple categories (e.g., explanation questions involving spatiotemporal reasoning or future prediction requiring domain expertise). Our current focus is on assigning a single primary category to each question to maintain clarity and focus.
> >  - That said, we have begun re-labeling the attributes of the dataset to allow each example to contain multiple categories where applicable. **We will provide an updated question distribution with these new labels as soon as possible (before the rebuttal period ends)**.
>
>
>
> `Question 1: The hexagon figure (Fig.4) is really difficult to interpret: I think this kind of figures makes sense if one compares 3 models at most. The score normalization makes things even worse: So we cannot see how well the best model performs in this way. I would definitely remove it from the paper, and put the Table 8 into the main text. Also note that the figure is embedded as "PNG/JPEG" instead of "PDF" (but removing it from the paper would make things much more better).`
>
> Many thanks for the great suggestion! We have now placed Table 8 into the main text and removed Figure 4 from the paper.
>
> We also want to clarify that all figures, including Figure 4, are embedded as PDFs. However, we have replaced unclear figures with more visually enhanced versions to improve clarity throughout the paper.
>
> ---
>
> `Question 2: Fig. 6 is also hard to interpret. I would rather group some selected models instead of error categories, so one can observe which model produces what kind of errors more or less.`
>
> Thank you for the suggestion. We have modified Figure 6 to group by models rather than error categories. This adjustment makes it easier to observe which models produce specific types of errors more or less. We have also at the same time increased the error sample sizes.
>
> ---
>
> `Question 3: Is there any example for auditory questions in the appendix? Can you please point? Can we see some synthetic questions/answer pairs generated using vision and audio side-by-side?`
>
> We have added examples of auditory questions in the appendix for easier reference. Additionally, we have included synthetic question-answer pairs generated side-by-side for vision and audio in the revised appendix.
>
> ---
>
> `Question 4: How much diverse is the language, i.e. questions and answers?`
>
> We want to reiterate that the questions and answers in the main subset of MMWorld are all annotated by human annotators so that to ensure the diversity of language.
>
> - The average question length is **11.39 words**.
> - The average answer length is **6.42 words**.
>
> To provide more details about language diversity, we have added the following statistics in the revision:
>
> - The vocabulary size (unique words) is **1,913** for questions and **2,292** for answers.
>
> These updates are now included in the revision.
>
> ---
>
> `Question 5: One suggestion: Please embed all figures as PDFs.`
>
> We want to clarify that all original figures in the paper are embedded in PDF format. However, we have replaced any unclear figures with clearer, higher-quality versions in the revised paper.
>
> ---
>
> `Question 6: It would be good to see many more examples in the appendix in a more compact format by avoiding large text with conversation bubbles.`
>
> Thank you again for your great suggestion!
> We have added more examples to the appendix in a more compact format, avoiding large text with conversation bubbles as suggested.

---

> ### Author Response · Authors · 2024-11-24
> **Additional Response to Reviewer ZqHq Regarding Weakness 4**
>
> Dear Reviewer ZqHq,
>
>
>
> We have completed the re-labeling of our dataset by assigning each QA to multiple categories. As a result, the subtask distribution has been updated accordingly, leading to a more balanced subtask distribution. The revised distribution is as follows: 22.1% in explanation, 13.7% in counterfactual thinking, 11.9% in future prediction, 16.5% in domain expertise, 14.1% in attribution understanding, 10.9% in temporal understanding, and 10.8% in procedure understanding.
>
> We have also uploaded a new revision to reflect these updates. Please refer to Figure 2 and our revision for further details. Do not hesitate to reach out if you have additional questions or require further clarification.
>
> Thank you for your insightful feedback and for giving us the opportunity to improve our work.
>
>
>
> Best,
>
> Authors

---

> ### Author Response · Authors · 2024-11-27
> **Looking forward to your response**
>
> Dear Reviewer ZqHq,
>
> Thank you for serving as a reviewer and providing valuable suggestions that have greatly helped us improve our work. We have carefully addressed your comments, enhanced our presentation, added more examples, refined the benchmark, and included new comparisons.
>
> We kindly invite you to revisit our paper in light of these updates. We hope our responses and improvements successfully address your concerns and encourage a reconsideration of your rating.
>
> Thank you again for your time and thoughtful feedback!
>
> Best,
>
> The Authors

---

> > ### Comment · Reviewer_ZqHq · 2024-11-27
> >
> > I want to increase my score to 6 after reading the author responses and checking the preprint. Most of my concerns are well addressed. That being said I would like to suggest that the authors should revisit Fig. 1 in the camera-ready version. I am specifically talking about the left part of the figure, where the smaller pie slices have low resolution and cannot be read easily.

---

> > > ### Author Response · Authors · 2024-11-27
> > >
> > > Thank you for taking the time to review our responses and our revisions, and increase your score!
> > >  We have updated Fig. 1 as per your feedback. Your suggestions have been greatly appreciated!

---

### Official Review · Reviewer_eShE · 2024-11-06

**Soundness:** 3
**Presentation:** 4
**Contribution:** 3
**Rating:** 6
**Confidence:** 4

**Summary:**

This paper proposed a comprehensive benchmark for MLLMs, covering 7 major disciplines, 69 subdisciplines, and multi-faceted reasoning tasks. The authors conducted experiments on 15 models (including both open-source and closed-source models) and analyzed the results.

**Strengths:**

* This benchmark encompasses 7 major disciplines and 69 subdisciplines, offering comprehensive coverage and highlighting its advantages over existing benchmarks.
* The authors introduced a new dataset, ensuring its quality through human annotation.
* The experiments are conducted on a large scale, covering mainstream MLLMs and providing analysis across different difficulty levels.

**Weaknesses:**

* Although the authors conduct an error analysis with six different categories, the error sample size is relatively small.
* The analysis of experimental results across different models lacks depth. For example, what specific differences exist between model and human reasoning? It would be helpful to compare with human performance, as in business discipline, the model achieves 90% accuracy, showing a strong understanding ability. It is not surprising that MLLMs perform better than average humans. It would be meaningful to see whether the model has reached a level close to or beyond human expert performance, especially in business discipline.

**Questions:**

1. Based on the results in Table 3, it appears that MLLMs perform significantly better in the Business discipline compared to other disciplines. Does this imply that the QA pairs in the Business discipline are relatively simpler, or does it indicate that MLLMs are stronger in the Business discipline than in other disciplines?
2. The authors emphasize that they have introduced a new dataset. I did not see an anonymous dataset link in the paper. Where can I find it?
3. Typos:
 • “an characteristic” in Figure 1.
 • “%%” at line 215.

---

> ### Author Response · Authors · 2024-11-22
> **Official Response to Reviewer eShE (1/2)**
>
> Thank you for your constructive feedback. Below, we address your comments and concerns point by point.
>
> ---
>
> ` Weakness 1: Although the authors conduct an error analysis with six different categories, the error sample size is relatively small.`
>
> We have expanded the error analysis to include 100 examples for each of the seven categories evaluated in the paper, increasing the sample size by tenfold compared to the original analysis. The results of this expanded analysis are presented in the revised version, along with an updated figure. The findings remain consistent with those from the previous analysis based on 10 examples, showing no significant differences in trends. This consistency reinforces the robustness of the original observations.
>
> ---
>
> ` Weakness 2: The analysis of experimental results across different models lacks depth. For example, what specific differences exist between model and human reasoning? It would be helpful to compare with human performance, as in business discipline, the model achieves 90% accuracy, showing a strong understanding ability. It is not surprising that MLLMs perform better than average humans. It would be meaningful to see whether the model has reached a level close to or beyond human expert performance, especially in business discipline.`
>
> As discussed in the main paper, MLLMs generally follow a trend where accuracy decreases as the difficulty level of questions increases, which aligns with human performance patterns. However, the correlation is not perfect, and there are notable differences in their capabilities:
>
>
> 1. **Strengths of MLLMs**:
>    - As highlighted in Figure 5b, models like GPT-4V can correctly answer expert-level questions that average humans often get wrong, particularly in disciplines such as Business and Health & Medicine. These areas are traditionally challenging for humans, indicating that MLLMs leverage data-driven insights effectively.
>
> 3. **Weaknesses of MLLMs**:
>    - MLLMs sometimes falter on simpler questions, likely due to a lack of contextual understanding or over-reliance on training data patterns. This is particularly evident in abstract disciplines like Art & Sports and Tech & Engineering, where human reasoning outperforms MLLMs. These discrepancies suggest differences in perception, cognition, and reasoning abilities between MLLMs and humans.
>
> 4. **Human Expert Performance**:
>    - The dataset was manually annotated by human experts, therefore human experts performance on the dataset would be close to 100%. This highlights a significant gap between human expert performance and MLLM performance. While models like GPT-4o achieve strong results in the business discipline (although there is still a 10% gap), they still fall short of human expert-level understanding in most areas.
>
> These findings indicate that MLLMs and humans can complement each other. MLLMs excel in providing data-driven insights, while human experts bring intuition and domain/contextual knowledge.

---

> ### Author Response · Authors · 2024-11-22
> **Official Response to Reviewer eShE (2/2)**
>
> ` Question 1: Based on the results in Table 3, it appears that MLLMs perform significantly better in the Business discipline compared to other disciplines. Does this imply that the QA pairs in the Business discipline are relatively simpler, or does it indicate that MLLMs are stronger in the Business discipline than in other disciplines?`
>
> We want to emphasize that the dataset has been carefully annotated and cross-checked by at least two professional researchers to ensure the quality and appropriate distribution of difficulty levels across all disciplines.
>
> To further analyze difficulty, we conducted human evaluation on the dataset using Amazon Mechanical Turk workers. The distribution of difficulty levels (Easy: Medium: Hard: Expert) based on average human performance (Easy = 3/3 correct answers; Medium = 2/3 correct; Hard = 1/3 correct; Expert = 0/3 correct) for each discipline is as follows:
>
> - **Art & Sports**: 16% : 35% : 30% : 19%
> - **Business**: 22% : 37% : 28% : 13%
> - **Science**: 21% : 34% : 29% : 16%
> - **Health & Medicine**: 22% : 38% : 28% : 12%
> - **Embodied Tasks**: 19% : 36% : 30% : 15%
> - **Tech & Engineering**: 11% : 35% : 33% : 21%
> - **Game**: 24% : 35% : 26% : 15%
>
> The difficulty level distribution is similar across all disciplines, reflecting the balanced nature of the dataset.
>
> We also now computed the overall average accuracy across all evaluated MLLMs (15 in total) for each discipline based on Table 3, and the results show a slight performance variation:
> - **Art & Sports**: 23.40%
> - **Business**: 38.88%
> - **Science**: 33.27%
> - **Health & Medicine**: 38.92%
> - **Embodied Tasks**: 29.22%
> - **Tech & Engineering**: 33.82%
> - **Game**: 34.33%
>
> While the differences are relatively small, MLLMs tend to perform slightly better on "Business" and "Health & Medicine." This may be because these disciplines often require domain-specific knowledge that is more readily available in public online datasets, which MLLMs leverage effectively during training.
>
> On the other hand, MLLMs perform relatively lower on "Art & Sports," "Embodied Tasks," and "Tech & Engineering." These disciplines require more complex perception, cognition, and reasoning abilities, as well as handling abstract concepts, which pose significant challenges for MLLMs.
>
> However, we would like to point out that while the overall trend indicates better performance in Business and Health & Medicine, this is not universal across all models. For instance, X-Instruct-BLIP-7B performs the worst on the Business discipline. This discrepancy can likely be attributed to the model’s pretraining datasets, such as COCO, VGQA, VQAv2, and ESC50, which contain limited data related to the Business domain. This highlights the impact of pretraining data on model performance across different disciplines.
>
>
> ---
>
> ` Question 2: The authors emphasize that they have introduced a new dataset. I did not see an anonymous dataset link in the paper. Where can I find it?`
>
> Thank you for your query. In our submitted supplements, we have included the JSON files from our dataset containing annotations and QA pairs. Additionally, we have now provided an anonymous Google Drive link that includes the videos in our dataset. You can access it here:
>
> [Anonymous Dataset Link](https://drive.google.com/file/d/1CUxRz6VoAYizvbNy69IOjNTCZ7zr6m9f/view?usp=sharing)
>
>
> ---
>
> ` Question 3: Typos: • “an characteristic” in Figure 1. • “%%” at line 215.`
>
> Thank you for your careful review. We have corrected the typo “an characteristic” in Figure 1 to “a characteristic” and removed the extraneous “%%” at line 215. These updates are reflected in the revised version of the paper.

---

> ### Author Response · Authors · 2024-11-25
> **Looking forward to your response**
>
> Dear Reviewer eShE,
>
> Thank you for your constructive feedback, which has greatly improved our work. We have addressed your comments, added deeper analysis of model versus human reasoning, expanded error analysis, included the anonymous dataset link, and corrected typos.
>
> We kindly invite you to revisit our paper in light of these updates and hope they address your concerns, supporting a reevaluation of your rating.
>
> Best,
>
> The Authors

---

> > ### Comment · Reviewer_eShE · 2024-11-29
> >
> > Thank you for your clarification and the revised version of the manuscript. After carefully reviewing the revision and considering the feedback provided by the other reviewers, I find the overall quality of the work has improved and I am slightly inclined to recommend accepting.

---

> > > ### Author Response · Authors · 2024-11-29
> > >
> > > Dear Reviewer eShE,
> > >
> > > Thank you for taking the time to review our clarifications and the revised manuscript. We are glad to hear that you are slightly inclined to recommend acceptance!
> > >
> > > If you have any additional feedback or suggestions, we would be happy to address them promptly. We would greatly appreciate it if you might consider raising your score!
> > >
> > > Best regards,
> > >
> > > The Authors

---

### Official Review · Reviewer_MwCy · 2024-11-09

**Soundness:** 2
**Presentation:** 2
**Contribution:** 2
**Rating:** 6
**Confidence:** 4

**Summary:**

This paper introduces MMWorld, a new benchmark for evaluating multimodal large language models' (MLLMs) ability to understand and reason about video content across multiple disciplines. The benchmark consists of 1,910 videos spanning 7 broad disciplines and 69 subdisciplines, with 6,627 question-answer pairs. The key innovation is its focus on multi-faceted reasoning (explanation, counterfactual thinking, future prediction, domain expertise) and multi-discipline coverage. The authors evaluate 15 MLLMs (both proprietary and open-source) on this benchmark and provide detailed analysis of model performance and error patterns.

**Strengths:**

1: The benchmark makes a novel contribution by filling an important gap in evaluating MLLMs' world modeling capabilities through video understanding across diverse disciplines.

2: The evaluation framework is comprehensive, covering seven broad disciplines and 69 subdisciplines while testing multiple reasoning types including explanation and counterfactual thinking.

3: The methodology demonstrates rigorous attention to detail, with well-documented data collection processes and multiple evaluation strategies.

4: The research provides valuable insights by identifying significant performance gaps in current MLLMs and revealing interesting differences between human and MLLM capabilities through detailed error analysis.

**Weaknesses:**

1. The dataset, while diverse, remains relatively similar compared to other modern benchmarks. Please include the recent works and discuss the difference between MMWorld and recent works, such as VideoMME, MMBench-Video.

2. The evaluation methodology shows potential bias through its heavy reliance on GPT-4V as a judge and in synthetic dataset generation.

3. The temporal reasoning aspect, though claimed as a key feature, could be more thoroughly explored and evaluated in the benchmark.

4. How to get the final results with the three subsets (Main Subset, Synthetic I, Synthetic II) in Table 3?

5. As the evaluated models cannot handle the audio modality, is the model's performance on the audio-related questions underestimated?

6. As the current VLMs are not the world model and the current questions in this benchmark are also limited to basic QA, limiting complex reasoning tasks. I think the author may reconsider the name of the benchmark to avoid over-claiming.

7. More in-depth analysis of the errors from the perspective of current VLM's architecture, training data, or training techniques would improve the work.

**Questions:**

As comments in Weaknesses

---

> ### Author Response · Authors · 2024-11-22
> **Official Response to Reviewer MwCy (1/3)**
>
> Thank you for your detailed review and constructive feedback. Below, we address your comments and questions point by point.
>
>
>
> `Weakness 1: The dataset, while diverse, remains relatively similar compared to other modern benchmarks. Please include the recent works and discuss the difference between MMWorld and recent works, such as VideoMME, MMBench-Video.`
>
> We appreciate your suggestion and have now included discussions comparing MMWorld with VideoMME and MMBench-Video in our revised paper. We would like to respectfully point out that these two works are concurrent with us and we highlight key differences below:
>
> - **VideoMME**: While VideoMME focuses on diversity in video types, it does not emphasize multi-discipline coverage like MMWorld. Our benchmark spans seven broad disciplines: *Art & Sports*, *Business*, *Science*, *Health & Medicine*, *Embodied Tasks*, *Tech & Engineering*, and *Games*, along with 69 subdisciplines. Additionally, MMWorld includes explanation and counterfactual reasoning within its multi-faceted reasoning tasks, which VideoMME lacks.
>
> - **MMBench-Video**: This benchmark focuses on fine-grained temporal reasoning in long video understanding. While temporal reasoning is part of MMWorld, our benchmark emphasizes a broader range of reasoning tasks and multi-discipline coverage. MMWorld includes temporal understanding questions as part of its multi-faceted reasoning approach.
>
> These two works have now been included in our revision.
>
> ---
>
> `Weakness 2: The evaluation methodology shows potential bias through its heavy reliance on GPT-4V as a judge and in synthetic dataset generation.`
>
>
> - Apart from GPT-4V, our paper also included evaluation methods using LLaVA, with the results shown in Table 9 of the Appendix. For clarity, we quote the table below:
>
> | Model                  | Accuracy (%) |
> |------------------------|--------------|
> | Video-Chat-7B          | 41.96        |
> | ChatUnivi-7B           | 39.81        |
> | mPLUG-Owl-7B           | 38.01        |
> | PandaGPT-7B            | 31.66        |
> | ImageBind-LLM-7B       | 31.65        |
> | X-Instruct-BLIP-7B     | 22.02        |
> | LWM-1M-JAX             | 16.81        |
> | Otter-7B               | 12.08        |
> | Video-LLaMA-2-13B      | 10.84        |
>
>
> - We also expanded our evaluation by incorporating rule-based metrics, where processed answers are compared to the ground truth for exact matching, using the LLMEvals toolkit. The results (%) are presented below:
>
> | Model                | Art & Sports | Business | Science | Health & Medicine | Embodied Tasks | Tech & Engineering | Game   | Average |
> |----------------------|--------------|----------|---------|-------------------|----------------|---------------------|--------|---------|
> | GPT-4o               | 38.10        | 74.64    | 59.48   | 68.84             | 52.17          | 63.82              | 63.68  | 50.63   |
> | Claude-3.5-Sonnet    | 43.44        | 52.31    | 48.25   | 45.03             | 25.69          | 49.74              | 47.13  | 43.99   |
> | GPT-4V               | 28.79        | 66.82    | 53.63   | 60.81             | 45.99          | 51.84              | 58.27  | 42.37   |
> | Gemini Pro           | 29.54        | 62.81    | 50.64   | 63.40             | 36.13          | 59.03              | 52.54  | 41.33   |
> | Video-LLaVA-7B       | 28.23        | 42.29    | 46.88   | 30.51             | 50.81          | 49.61              | 39.06  | 38.45   |
> | Video-Chat-7B        | 31.09        | 42.15    | 25.67   | 43.18             | 32.64          | 33.56              | 35.79  | 34.18   |
> | ChatUnivi-7B         | 19.22        | 50.30    | 43.32   | 57.36             | 37.16          | 48.44              | 41.83  | 33.85   |
> | mPLUG-Owl-7B         | 22.92        | 53.09    | 39.52   | 56.47             | 19.10          | 35.72              | 49.39  | 33.65   |
> | Video-ChatGPT-7B     | 21.10        | 32.44    | 30.38   | 48.96             | 29.36          | 35.47              | 29.01  | 30.88   |
> | PandaGPT-7B          | 19.91        | 35.41    | 32.88   | 35.53             | 28.39          | 35.74              | 31.83  | 29.91   |
> | ImageBind-LLM-7B     | 19.52        | 35.41    | 26.80   | 27.85             | 37.49          | 35.47              | 32.79  | 27.62   |
> | X-Instruct-BLIP-7B   | 16.55        | 13.16    | 18.79   | 26.25             | 14.76          | 19.10              | 20.71  | 18.14   |
> | LWM-1M-JAX           | 9.38         | 14.47    | 12.97   | 18.53             | 21.13          | 17.55              | 9.12   | 14.01   |
> | Video-LLaMA-2-13B    | 4.83         | 17.56    | 18.27   | 28.17             | 12.34          | 15.02              | 19.30  | 13.29   |
> | Otter-7B             | 13.43        | 15.42    | 7.77    | 6.22              | 10.78          | 12.48              | 11.75  | 11.53   |

---

> ### Author Response · Authors · 2024-11-22
> **Official Response to Reviewer MwCy (2/3)**
>
> While the performance is slightly lower when using rule-based metrics compared to GPT-4V as a judge, the rankings remain almost consistent, demonstrating reliability across different evaluation methods.
>
> Meanwhile, we emphasize that, as mentioned in the main paper, the effectiveness of using GPT-4V as a judge has been widely validated in recent works ([1], [2], [3], [4]). For example, the recent work MMBench-Video you mentioned also uses GPT-4 as a judge.
>
> We have also compared GPT-4-based evaluation with human evaluators and validated its effectiveness in our paper in Appendix D.2, showing an error rate of only 4.76% across 189 examples. This demonstrates its reliability as an evaluator.
>
> Finally, for synthetic data generation, this approach aligns with practices in prior works ([5], [6]). Although GPT-4V was used, the generated data was carefully reviewed and validated by at least two professional researchers to ensure correctness and accuracy. Additionally, Amazon Mechanical Turk workers and college students were also asked to solve those problems in our dataset so as to validate them further.
>
>
> **References:**
>
> [1] Video-chatGPT: Towards detailed video understanding via large vision and language models
> [2] GPT-4 as an effective zero-shot evaluator for scientific figure captions
> [3] Is GPT-4 a reliable rater? Evaluating consistency in GPT-4 text ratings
> [4] GPT-Eval: NLG evaluation using GPT-4 with better human alignment
> [5] ShareGPT4Video: Improving Video Understanding and Generation with Better Captions
> [6] Video Instruction Tuning with Synthetic Data
>
> ---
>
> ` Weakness 3: The temporal reasoning aspect, though claimed as a key feature, could be more thoroughly explored and evaluated in the benchmark.`
>
> To address this, we have added two additional experiments for the temporal reasoning questions:
>
> 1. **Reduced Video Frames**: Videos were processed by reducing the number of frames to 1/5 of the original, testing the models' ability to reason with limited temporal information.
>
> 2. **Shuffled Video Frames**: Videos were processed by shuffling their frames, testing the models' ability to reason when temporal order is disrupted.
>
> The evaluated results are as follows:
>
> | Models               | GPT-4o | GPT-4V | Claude-3.5-Sonnet | Gemini Pro | Video-LLaVA | Video-Chat-7B | Video-ChatGPT-7B | ImageBind-LLM-7B | PandaGPT-7B | ChatUnivi-7B | Video-LLaMA-2-13B | X-Instruct-BLIP-7B | LWM-1M-JAX | Otter-7B | mPLUG-Owl-7B |
> |----------------------|--------|--------|--------------------|------------|-------------|----------------|-------------------|------------------|-------------|---------------|---------------------|-------------------|-----------|----------|---------------|
> | Original             | 40.90  | 27.17  | 25.77             | 24.65      | 34.45       | 25.77          | 23.53             | 19.89           | 28.01       | 22.97         | 6.16                | 11.20             | 7.00      | 9.52     | 20.17         |
> | Shuffled Videos      | 35.11  | 22.04  | 21.58             | 20.19      | 18.47       | 21.50          | 21.62             | 16.19           | 24.35       | 19.41         | 5.02                | 9.88              | 5.75      | 3.25     | 18.19         |
> | Less Video Frames    | 32.19  | 22.33  | 19.45             | 18.97      | 28.50       | 20.19          | 20.17             | 14.98           | 22.57       | 17.14         | 4.58                | 8.95              | 5.56      | 7.93     | 16.59         |
>
> From the results, we observe significant performance drops for both reduced and shuffled video frames compared to the original videos. This underscores the importance of temporal coherence and sufficient temporal information for reasoning tasks. We will add those results into the revision.
>
> ---
> ` Weakness 4: How to get the final results with the three subsets (Main Subset, Synthetic I, Synthetic II) in Table 3?`
>
> We would like to clarify that the results in Table 3 correspond to the *Main Subset*, which is fully human-annotated. The *Synthetic I* and *Synthetic II* subsets, which focus on ablation studies of individual modalities (e.g., audio-only or video-only perception), are reported separately in Table 4.
>
> The reason for keeping these results distinct is that MMWorld consists of two primary components:
>
> 1. **Main Subset**: This subset evaluates multi-discipline and multi-faceted reasoning abilities of the models comprehensively. It is designed to assess the models' integrated performance across different disciplines.
>
> 2. **Synthetic Subsets**: These subsets are crafted to isolate and analyze specific modality-related capabilities (e.g., visual or auditory perception), towards world modeling
>
> By separating the results, we ensure that each subset's specific evaluation goal is appropriately highlighted. This separation allows us to maintain clarity and perform better analysis in the interpretation of model performance.

---

> ### Author Response · Authors · 2024-11-22
> **Official Response to Reviewer MwCy (3/3)**
>
> ` Weakness 5: As the evaluated models cannot handle the audio modality, is the model's performance on the audio-related questions underestimated?`
>
> We would like to clarify that the *Main Subset* is not specifically designed to measure the ability to answer audio-related questions.
>
> In our benchmark, we evaluate models separately for audio and visual modalities using two dedicated subsets: *Synthetic I* (audio-only) and *Synthetic II* (visual-only). These subsets enable comprehensive testing of the models' performance on individual modalities, ensuring that audio and visual components are assessed independently and fairly.
>
> ---
>
> ` Weakness 6: As the current VLMs are not the world model and the current questions in this benchmark are also limited to basic QA, limiting complex reasoning tasks. I think the author may reconsider the name of the benchmark to avoid over-claiming`
>
> As discussed in the main paper, while current VLMs are not yet world models, we believe that “overcoming the unique challenges presented in MMWorld is essential and necessary towards comprehensive world modeling.” The MMWorld benchmark is “designed to rigorously evaluate the capabilities of Multimodal Large Language Models (MLLMs) in world modeling,” which is why it was originally named MMWorld.
>
> Regarding the task format, we adopted QA tasks in MMWorld because they are easier to quantify and evaluate objectively. Open-ended generation tasks pose challenges due to their subjective nature, whereas QA tasks allow for more straightforward and reliable assessments. Using multiple-choice questions is a well-established evaluation method for multimodal video understanding tasks, especially when complex reasoning is required.  Additionally, the questions in MMWorld are carefully designed to target specific aspects of complex reasoning, such as temporal understanding and other reasoning facets. This approach enables focused and controlled evaluation with a limited set of candidate answers, ensuring clarity and precision in assessing model capabilities.
>
> Nevertheless, to address concerns of potential over-claiming, we have renamed the benchmark to **MMVU: Towards Multi-discipline Multi-faceted Video Understanding Evaluation** and submitted a revised version reflecting this change.
>
> ---
>
> ` Weakness 7: More in-depth analysis of the errors from the perspective of current VLM's architecture, training data, or training techniques would improve the work.`
>
> Thank you for the suggestion. In response, we have increased the sample size for error analysis to 100 examples and added more detailed insights in the revised paper. Below, we summarize key findings:
>
> - Video-LLaVA demonstrates fewer errors across all types compared to other open-source MLLMs. This superior performance can be attributed to its extensive training on high temporal resolution video data, which includes 558K LAION-CCSBU image-text pairs and 702K video-text pairs from WebVid. It's relative low **Visual Perception Errors (PE)** is likely due to the adoption of the CLIP ViT-L/14 vision model in LanguageBind. It also show relative **Lower Hallucination Errors (HE) and Reasoning Errors (RE)**, which may be attributable to the large volume of image-video-text pairings in its training data.
>
> - mPLUG-Owl exhibits higher visual perception errors, potentially due to its use of a weaker video embedder, such as the TimeSformer, which may be less effective for complex visual reasoning tasks.
>
> - VideoChat vs. Video-LLaMA: Although these models share similar architectures, VideoChat incorporates a GMHRA module for temporal aggregation instead of the video QFormer used in Video-LLaMA. This difference leads to overall fewer errors in VideoChat, suggesting that an effective temporal aggregation module plays a critical role in improving performance.
>
> These findings highlight the significant impact of training data, architectural choices, and temporal resolution on the error rates and reasoning capabilities of current VLMs. Detailed results and analysis are provided in the revised manuscript.
>
> ---
>
> We hope these updates address your concerns, and we appreciate your thoughtful feedback in improving the paper.

---

> ### Author Response · Authors · 2024-11-25
> **Looking forward to your response!**
>
> Dear Reviewer MwCy,
>
> We greatly appreciate your feedback, which has been invaluable in improving the clarity and quality of our work. We have carefully addressed your comments, clarified potential misunderstandings, and enhanced the paper, including additional experiments on temporal reasoning, comparisons with recent works (e.g., VideoMME, MMBench-Video), and in-depth error analysis.
>
> We kindly invite you to revisit our paper in light of these updates and clarifications. We hope these changes address your concerns and would greatly appreciate it if you could consider whether they warrant a reevaluation of your rating.
>
> Best,
>
> The Authors

---

> > ### Comment · Reviewer_MwCy · 2024-11-25
> >
> > Thanks for the detailed response, I will raise my score to 6.

---

> > > ### Author Response · Authors · 2024-11-25
> > >
> > > Thank you for taking the time to review our responses and revisions, which have helped improve our work. We also greatly appreciate your increased rating!

---

### Meta-Review · Area_Chair_t3B7 · 2024-12-16

**Metareview:**

This paper proposes a world model evaluation in videos. All the reviews are positive in the final review. The raised concerns have been addressed in the authors' responses and acknowledged by the reviewers. Overall, the AC has checked all the files and stands on the reviewers' side. The proposed evaluation brings metrics to the current foundation model community. The authors shall incorporate all the revisions suggested by reviewers in the camera-ready version.

**Additional Comments On Reviewer Discussion:**

Reviewers have raised multiple issues regarding the paper presentation, technical setting, and experiments. These aspects are well addressed by the authors, and are acknowledged by the reviewers.

---

### Decision · Program_Chairs · 2025-01-22

Accept (Poster)